



# Aerosol effective radius governs the relationship between cloud condensation nuclei (CCN) concentration and aerosol backscatter

Emily Lenhardt[1], Lan Gao[1], Chris A. Hostetler[2], Richard A. Ferrare[2], Sharon P. Burton[2], Richard H. Moore[2], Luke D. Ziemba[2], Ewan Crosbie[2,3], Armin Sorooshian[4,5], Cassidy Soloff[4,5], Jens Redemann[1]

[1]School of Meteorology, University of Oklahoma, Norman, OK, 73072, United States
[2]NASA Langley Research Center, Hampton, VA, 23681, United States
[3]Analytical Mechanics Associates, Hampton, VA, 23666, United States
[4]Department of Hydrology and Atmospheric Sciences, University of Arizona, Tucson, AZ 85721, United States
[5]Department of Chemical and Environmental Engineering, University of Arizona, Tucson, AZ 85721, United States

*Correspondence to*: Emily D. Lenhardt (emily.lenhardt@ou.edu)

**Abstract.** Understanding the vertical distribution of cloud condensation nuclei (CCN) concentrations is crucial for reducing uncertainty associated with aerosol-cloud interactions (ACI) and their effective radiative forcing (ERF$_{aci}$). Many studies take advantage of widely available remote sensing observations to develop proxies, parameterizations, and relationships between CCN concentration and aerosol optical properties (AOPs). Such methods generally provide a good constraint for CCN concentration, but many uncertainties and limitations exist, generally related to high relative humidity (RH), environments with internal or external mixtures of several different aerosol types, and differences in parts of the aerosol size distribution relevant for both CCN and AOPs. In this study we use in situ observations of the aerosol size distribution and chemical composition in a recent airborne field campaign to inform theoretical calculations of CCN concentration (CCN$_{theory}$) and aerosol backscatter at 532 nm (BSC$_{theory}$) with the purpose of understanding the dominant governing factors of the CCN$_{theory}$ - BSC$_{theory}$ relationship. Estimates from random forest models indicate that for smoke, marine, and urban aerosols the aerosol size distribution, as parameterized by effective radius (R$_{eff}$), is the most important predictor of the CCN$_{theory}$ − BSC$_{theory}$ relationship. We further investigate how R$_{eff}$ impacts CCN$_{theory}$:BSC$_{theory}$ and find an exponential relationship between the parameters. We find that modelling CCN$_{theory}$:BSC$_{theory}$ using this exponential R$_{eff}$ relationship can explain about 68-79% of the variance in the CCN$_{theory}$ - BSC$_{theory}$ relationship. These findings suggest that including information about aerosol size is critical for future studies in constraining CCN concentration from AOPs.

## 1 Introduction

Natural and anthropogenic atmospheric aerosols and their interactions with clouds and radiation have a significant role in climate change and uncertainties in future climate predictions. Specifically, the highest uncertainties compared to other climate forcings are attributed to effective radiative forcing due to interactions between aerosols and clouds (ERF$_{aci}$; Forster et al., 2021). Much of the uncertainty of aerosol-cloud interactions (ACI) is due to limited process-level understanding (Boucher et al., 2013) and observing methods. For example, there is limited ability for passive satellite instruments to retrieve cloud and





aerosol properties simultaneously in the same environment. Hygroscopic aerosol growth in high relative humidity (RH) environments can also complicate observations (Rosenfeld et al., 2014). Additionally, varying observational scales and meteorological conditions may buffer the responses of clouds to aerosol perturbations (Stevens & Feingold, 2009).

Untangling the impact of ACI from such observational complications requires information on the distribution of those aerosols that interact with clouds by nucleating cloud droplets, i.e., cloud condensation nuclei (CCN). More specifically, knowledge of the vertical distribution of CCN concentration relative to clouds is needed to properly assess and understand ACI. The main challenge in understanding the vertical distribution of CCN lies in the sparsity of in situ observations. Ground-based observations are useful in terms of the length of available observations, but they lack vertical extent. Alternatively,

aircraft-based observations can provide observations of CCN closer to cloud base over shorter campaign periods, but these observations are expensive and less frequently available. Therefore, many studies have developed parameterizations, proxies, and retrieval methods to determine CCN from more commonly available remotely sensed observations of aerosol optical properties (AOPs).

One of the most widely used proxies for CCN concentration is aerosol optical depth (AOD), a column-integrated

measure of aerosol extinction (EXT). While AOD may approximate CCN concentration over large spatiotemporal extents (Stier, 2016), it often cannot fully explain CCN variance (Andreae, 2009; Shinozuka et al., 2015; Stier, 2016; Choudhury & Tesche, 2022a; Choudhury & Tesche, 2022b), lacks any information about the vertical distribution of CCN, and is subject to effects of aerosol swelling and cloud contamination (Rosenfeld et al. 2016; Patel et al. 2024). Several studies have related CCN to a combination of other AOPs from lidar and satellite such as aerosol extinction, scattering and backscattering

coefficients, backscatter fraction, or the ratio of backscattering to total scattering, single scattering albedo (SSA), scattering Ångström exponent, and aerosol index (AI), which is the product of Ångström exponent and extinction (Ghan & Collins, 2004; Ghan et al., 2006; Kapustin et al., 2006; Shinozuka et al. 2009; Jefferson, 2010; Liu & Li, 2014; Shinozuka et al., 2015; Mamouri & Ansmann, 2016; Stier, 2016; Tskeri et al., 2017; Lv et al., 2018; Shen et al., 2019; Choudhury & Tesche, 2022a; Choudhury & Tesche, 2022b; Lenhardt et al., 2023; Patel et al., 2024; Redemann & Gao, 2024). Among such approaches,

AOPs can provide constraints for CCN but several underlying uncertainties and limitations exist.

One fundamental issue in relating CCN concentration to AOPs is that particles that act as CCN are generally smaller than particles that have a more significant impact on AOPs when measured at visible wavelengths. Most CCN fall in the Aitken and accumulation modes of the aerosol size distribution, and studies have shown that changes in the aerosol size distribution are the primary drivers of changes in the CCN spectrum (Dusek et al., 2005; Miao et al., 2015; Perkins et al., 2022). In terms

of AOPs, many are dominated by coarse mode particles (Shinozuka et al., 2015) and optical measurements tend to be insensitive to small particles that activate as CCN (Jefferson, 2010), causing further uncertainty in correlating both measurements. Another common issue in relating CCN to AOPs is hygroscopic growth of aerosols at high ambient RH. Hygroscopic growth increases aerosol size, thus increasing their light scattering. However, the lack of a corresponding increase in CCN concentration (Shinozuka et al., 2015) causes CCN – AOP relationships to change rapidly at high RH (Liu & Li, 2014;

Shinozuka et al., 2015; Stier, 2016; Wang et al., 2025). Since CCN are of particular interest in humid environments near cloud



base, this issue can become problematic for ACI applicability. Additionally, aerosol chemical composition influences both CCN concentration and AOPs and their relationship. Some studies have found that CCN – AOP relationships are more uncertain for observations of marine aerosols (Liu & Li, 2014; Shen et al., 2019), which may be related to their more dominant coarse mode and tendency for marine aerosol shapes to be non-spherical (Fitzgerald, 1990; von Hoyningen-Huene & Posse,

1996; Bi et al. 2018). In summary, the three most common sources of potential error when relating CCN to AOPs are related to high ambient RH, the shape of the aerosol size distribution, and aerosol chemical composition.

While each of these sources of uncertainty and potentially weak correlation have been noted by numerous studies, many that investigate underlying causes of error focus on each source individually. Additionally, many studies that take a modeling or calculation-based approach to investigating CCN and/or AOPs often use idealized, generated, or average size

distributions as a starting point (Li et al., 2015; Lowe et al., 2016; Shen et al., 2019) or vary individual observed size distributions in concentration but not the functional shape (Chuang et al., 1999). While this approach avoids the uncertainties inherent to in situ aerosol size distribution observations, it also does not capture the full range of variability seen in observed size distributions.

In this study we investigate the collective impact of ambient RH, aerosol size distribution, and aerosol chemical

composition on CCN – AOP relationships using a broad range of actual observed aerosol properties. Specifically, we follow and expand on Lenhardt et al. (2023), hereafter L23, by applying the same methodology to multiple aerosol types under a variety of ambient RH conditions observed during the Aerosol Cloud meTeorology Interactions oVer the western ATlantic Experiment (ACTIVATE) campaign (Sect. 2). L23 focused on optimizing a linear regression model between in situ CCN concentration and aerosol extinction and backscatter from the High Spectral Resolution 2 (HSRL-2) for observations of smoke

in mostly dry (RH ≤ 50%) conditions during the ObseRvations of Aerosols above CLouds and their intEractionS (ORACLES) campaign. In this study, we perform an L23-motivated analysis and follow it with a more in-depth investigation of the underlying factors that govern how CCN concentration and backscatter at 532 nm (BSC) are related to understand which one may be the most important predictor. To achieve this, we perform observation-informed theoretical calculations of CCN ($CCN_{theory}$) and aerosol BSC ($BSC_{theory}$) using in situ observed aerosol size distribution and chemical composition as inputs to

both the κ-Köhler and Mie theories (Sect. 3 and 4). Throughout the study, all observations and calculations of BSC will be at 532 nm. Requisite input data for both calculations comes from ACTIVATE in situ observations, meaning that we do not assume average values for the hygroscopicity parameter or use a singular idealized, representative aerosol size distribution. This approach allows us to capture the observation-informed variability in aerosol size and composition to investigate how such variability impacts the theoretical relationship between CCN and BSC.

**2 Field Deployment Background and Motivation for the Present Study**

The National Aeronautics and Space Administration (NASA) ACTIVATE campaign took place between February 2020 and June 2022 across six deployments over the northwest Atlantic Ocean and generated a unique in situ and remote sensing data





set relevant for investigating aerosol-cloud-meteorology interactions. Unlike subtropical regions frequently chosen for ACI-related campaigns, the northwest Atlantic features numerous cloud types, including warm and mixed phase cumulus, that are

less well-understood than stratocumulus cloud decks (Sorooshian et al., 2023). Additionally, observations over different seasons allow for analysis of a wide range of aerosol and meteorological conditions. Data were collected using coordinated flights of the NASA Langley Research Center HU-25 Falcon for in situ measurements and King Air aircraft for remote sensing observations. In this study, we collocate in situ and remote sensing observations from both aircraft. The study region and locations of these collocated data points broken down by aerosol type are shown in Fig. 1.


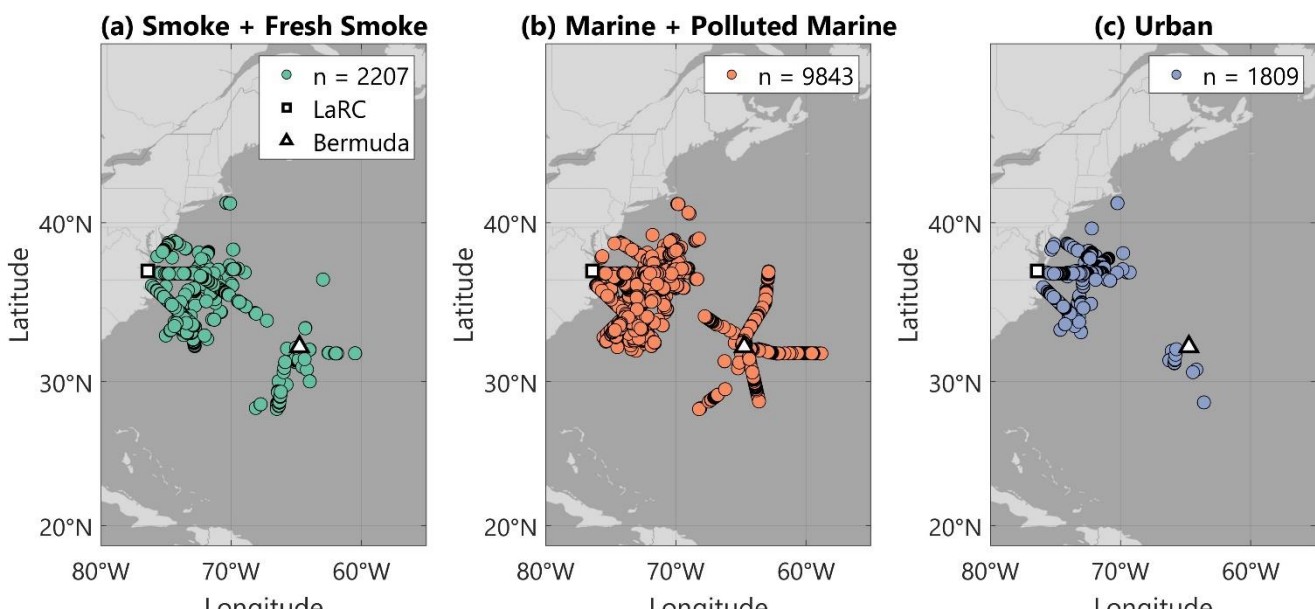

**Figure 1: Maps showing the ACTIVATE study area and locations of collocated data points for observations of (a) smoke and fresh smoke, (b) marine and polluted marine, and (c) urban aerosols. Langley Research Center (LaRC) in Hampton, Virginia, and Bermuda, the two major bases of operations, are also shown.**


During the ACTIVATE campaign, the HU-25 Falcon aircraft conducted profiling flights within, above, and below boundary layer clouds while collecting in situ observations, and the spatially coordinated King Air flew above the Falcon (~9 km) conducting remote sensing observations and launching dropsondes (Sorooshian et al., 2023). Following the methodology of L23, our first step is a direct comparison between observed CCN concentration (CCN_obs) and BSC at 532 nm (BSC_obs), the

instrumentation for which are described in Section 3.1. Due to the different spatiotemporal resolutions of the in situ and remote sensing data sets, we first collocate both data sets to enable a one-to-one comparison. Fortunately, ACTIVATE prioritized systematic and spatially coordinated flights between both aircraft, with approximately 73% of the cumulative dataset having the two aircraft within 6 km and 5 minutes of one another (Schlosser et al., 2024). Therefore, collocation between both aircraft



results in many collocated data points for remaining analyses. Our collocation process uses three independent collocation
criteria to find in situ measurements that fall within a set amount of time (dt = ±0.1 h) from when an HSRL-2 profile was
measured, within a set horizontal distance (dd = ± 0.01°, or ± ~1.1 km) from the profile, and within set vertical bins (dh = 45
m). After these criteria have been applied, in situ observations that meet all three criteria are averaged to enable a one-to-one
comparison with HSRL-2 BSC. For more details on our sensitivity testing method to determine appropriate values for dt, dd,
and dh, and a schematic describing the collocation process, see L23.

We analyze the correlation between collocated $CCN_{obs}$ and $BSC_{obs}$ separated by aerosol type (Fig. 2), indicated by
the HSRL-2 Aerosol ID product (Sect. 3.1.2). Following L23, we fit all relationships using a bisector regression to account for
both variables being measured with observational uncertainty. Additionally, we show $R^2$, root mean square error (RMSE), and
number of data points (n). Observations are limited to a small supersaturation range of 0.36-0.38% and marker colors
correspond to ambient RH. One of our primary findings in L23 was that the correlation between $CCN_{obs}$ and $BSC_{obs}$ was
strongest at low ambient RH (≤ 50%). Therefore, we show separate statistics and regression lines for all observations and the
subset observed at RH ≤ 50%.

$R^2$ values for all aerosol types across the full RH spectrum range from 0.0014-0.14, and for RH ≤ 50% range from
0.0023-0.038. For all RHs, the correlation is strongest for URB, while smoke has the highest correlation under limited RH
conditions. For SFS and URB, $R^2$ decreases when limiting the data set to low RH, contrary to the findings of L23. In the case
of the SFS and MPM analyses, RMSE increases when limiting the data set to low RH. Overall, RMSE varies from 342-541
$cm^{-3}$, and these values are significantly higher than the median $CCN_{obs}$ uncertainty of approximately 150 $cm^{-3}$ for this data set
(assuming a relative uncertainty of 10% as reported in the data). Additionally, we see the impact of hygroscopic growth most
clearly in the MPM results, where several observations made at RH > 80% show increased $BSC_{obs}$ associated with nearly
constant, and low, $CCN_{obs}$ values. This aerosol type is primarily influenced by sea salt, one of the most hygroscopic aerosols
with a high growth factor and kappa that can range from 0.91-1.33 (Petters & Kreidenweis, 2007).



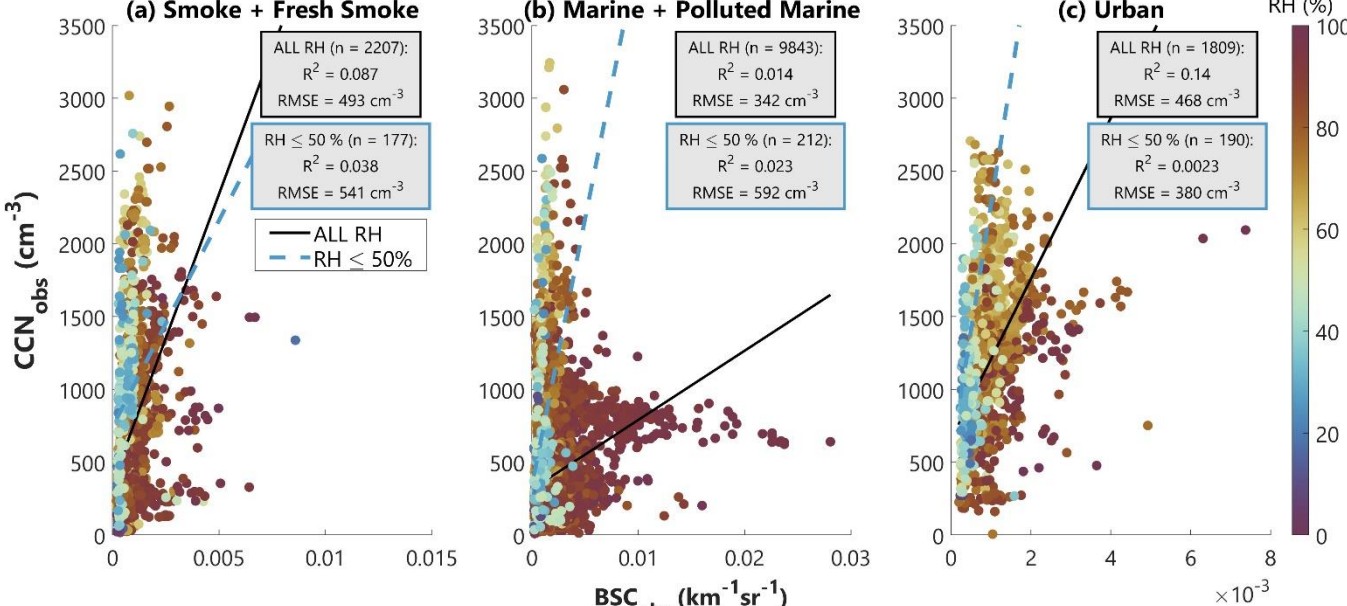

**Figure 2: Bisector regression of CCN$_{obs}$ vs. BSC$_{obs}$ for (a) smoke and fresh smoke, (b) marine and polluted marine, and (c) urban aerosols. This combined data set covers all years of ACTIVATE and represents 76 flight days. Supersaturation for these observations ranges between 0.36-0.38%. The number of collocated data points (n) is given, as well as the R$^2$ value and root mean square error (RMSE). Statistics are given for the full data set (black-outlined box), as well as the subset of data observed at RH ≤ 50% (blue-outlined box). The solid black line of best fit applies to the full data set, and the dashed blue line of best fit applies to the low RH subset of data.**

Unlike in L23, the direct relationship between CCN$_{obs}$ and BSC$_{obs}$ in the ACTIVATE data cannot be represented well using a linear approximation. We find that even when limiting the data set to observations made at low ambient RH that the correlation is weak and scatter around the regression line is high. Another difference between this analysis and ORACLES results is the relatively low frequency of observations made in low RH environments. While more than half of the smoke plume observations in ORACLES were made at low RH, only about 2-10% of the observations in Fig. 2 were observed at low 155    RH since the HU-25 Falcon primarily sampled in the marine boundary layer (MBL) during ACTIVATE. This observed non-linearity between CCN$_{obs}$ and BSC$_{obs}$ in the ACTIVATE data (Fig. 2) serves as motivation for the rest of this study – unlike L23, we do not try to optimize a linear relationship between CCN$_{obs}$ and BSC$_{obs}$. Rather, we perform calculations of CCN$_{theory}$ and BSC$_{theory}$ based on actual observations of aerosol size distribution and chemical composition to understand this observed non-linearity and determine which factors dominate in governing the CCN$_{theory}$ – BSC$_{theory}$ relationship. The goal of this 160    theoretical investigation is to use observations from ACTIVATE as a basis to determine what additional information is most important in constraining CCN concentration from remotely sensed AOPs such as lidar aerosol backscatter.





## 3 Data & Theoretical Calculation Methods

The four primary ACTIVATE data sets used in this study are described in Sect. 3.1 and summarized in Table 1. Our methodologies for calculations of $CCN_{theory}$ and $BSC_{theory}$ are outlined in Sect. 3.2 and 3.3, respectively. Lastly, we describe

pre-analysis data filtering steps in Sect. 3.4.

### 3.1 Instrumentation

### 3.1.1 Droplet Measurement Technologies (DMT) CCN Counter

The Droplet Measurement Technologies (DMT) CCN counter measures in situ CCN concentration at multiple levels of water vapor supersaturation (S) and can be run in constant S or scanning S modes (Moore & Nenes, 2009), with most observations

from ACTIVATE made at approximately S = 0.37%. This instrument is designed as a continuous-flow streamwise thermal-gradient chamber (CFSTGC; Roberts & Nenes, 2005) where a quasi-uniform supersaturation is generated in the center of a cylindrical flow chamber as heat and water vapor are continuously transported from wetted walls under a temperature gradient. Supersaturation levels vary based on the instrument pressure, flow rate, and imposed column temperature gradient. The continuous-flow feature enables quick (1 Hz frequency) sampling (Roberts & Nenes, 2005), which is important for airborne

observations in quickly evolving environments. At the end of the growth chamber, aerosols that activated into droplets with a radius greater than 0.5 μm are counted as CCN. The uncertainty reported for CCN concentration is ±10%, with a supersaturation uncertainty of ±0.04% (Rose et al., 2008).

### 3.1.2 High Spectral Resolution Lidar 2 (HSRL-2)

The NASA Langley Research Center HSRL-2 measures aerosol backscatter and depolarization at 355, 532, and 1064 nm and

aerosol extinction via the HSRL technique at 355 and 532 nm (Shipley et al., 1983; Hair et al., 2008; Burton et al., 2018). Using the spectral distribution of the return signal, the HSRL measurement technique enables separation of aerosol and molecular backscatter signals, which in turn allows independent, accurate retrieval of aerosol backscatter and extinction profiles without reliance on external assumptions such as the value of the lidar ratio, as is common for basic elastic backscatter lidars (Hair et al., 2008). In this study, we focus on particulate backscatter at 532 nm. The 532 nm wavelength is more

frequently available in the data set, and results from L23 suggested that when directly relating CCN concentration with HSRL-2 backscatter and extinction at 355 and 532 nm, there was no substantial difference in performance between either product or wavelength. Additionally, BSC at 532 nm is broadly applicable to existing ground-based and spaceborne lidars and may also be more applicable to observations from a Raman lidar potentially included in the future NASA Atmosphere Observing System (AOS) mission. Uncertainty in the HSRL-2 observables depends on factors such as contrast ratio and aerosol loading, but

uncertainties within 5% can be achieved under certain conditions (Burton et al., 2018).

Additionally, since we are interested in the impact of different aerosol types on the CCN – BSC relationship, we also use the HSRL-2 Aerosol ID product. This Aerosol ID is a qualitative indication of aerosol type from a classification scheme





based on HSRL-2 measurements of aerosol intensive parameters including lidar ratio at 532 nm, 1064-to-532 nm backscatter color ratio, depolarization at 532 nm, and depolarization spectral ratio (Burton et al., 2012). The method categorizes eight

particle types, which include ice, dusty mix, marine, urban/pollution, smoke, fresh smoke, polluted marine, and dust. In this study, we combine smoke with fresh smoke (SFS) and marine with polluted marine (MPM) due to similarity in their optical properties. We also consider the urban/pollution (URB) aerosol type.

### 3.1.3 Scanning Mobility Particle Sizer (SMPS) and Laser Aerosol Spectrometer (LAS)

In situ aerosol size distributions come from a combination of the Scanning Mobility Particle Sizer (SMPS) and Laser Aerosol

Spectrometer (LAS), both part of the Langley Aerosol Research Group Experiment (LARGE) instrument suite. The uncertainty reported for data from the SMPS and LAS in ACTIVATE is 20% (Sorooshian et al., 2023).

The SMPS uses a soft X-ray aerosol charger (TSI model 3088) to impart an aerosol sample with a known charge distribution and classifies the electric mobility of charged particles with a nano-column differential mobility analyzer (DMA; TSI Model 3085). The particle concentration of aerosols between 0.003-0.089 μm midpoint diameter is then measured using

an ultrafine condensation particle counter (CPC; TSI Model 3776; Moore et al., 2017). The resultant size-resolved particle number size distribution is reported as $dN/dlogD_p$ at standard temperature and pressure (STP; 0°C and 1013.25 mb) with 45 second time resolution. Size-dependent corrections have been applied based on laboratory calibration that result in excellent closure with total number concentrations measured by an independent CPC (Sorooshian et al., 2023).

The LAS measures the particle number size distribution ($dN/dlogD_p$) of aerosols with midpoint diameters between

0.1-3.5 μm using an optical method where light intensity scattered from a laser is used to measure particle size (TSI Model 3340; Moore et al., 2021). Unlike less sophisticated optical instruments, a wide-angle scattering technique allows for a monotonic response to the intensity of light scattering to resolve Mie scatter sizing issues. Additionally, an intracavity helium-neon laser design allows for higher light scattering sensitivity at lower laser power. The LAS is calibrated with monodisperse ammonium sulfate particles owing to a refractive index (n = 1.52) close to many ambient aerosols (Shingler et al., 2016).

Concentrations are reported at STP and with 1 Hz time response. The combination of SMPS and LAS measurements provides a continuous size distribution.

### 3.1.4 Aerodyne High-Resolution Time-of-Flight Aerosol Mass Spectrometer (AMS)

The Aerodyne high-resolution time-of-flight (HR-ToF) aerosol mass spectrometer (AMS) measures submicron, non-refractory composition, including mass concentrations of sulfate, nitrate, ammonium chloride, and organic aerosols, as well as several

mass spectral markers (DeCarlo et al., 2008; Sorooshian et al., 2023). The AMS uses an aerodynamic lens to focus particles into a narrow beam within a vacuum chamber, and particles are then impacted onto a 600°C vaporizer. This results in flash vaporization and ionization of non-refractory aerosol components. Ion extraction then allows for the generation of a complete mass spectrum (Jimenez et al., 2003; Drewnick et al. 2005). Refractory components including black carbon, sea salts, and crustal species are not measured efficiently by the AMS (Jimenez et al., 2003; Cai et al., 2018). Additionally, AMS





measurements apply to aerosols with an aerodynamic diameter of approximately 60-600 nm, where transmission efficiencies can be nearly 100% (Jimenez et al., 2003). Although this size range does not cover the full aerosol size distribution, it covers sizes that make up the majority of CCN, so uncertainty due to particle sizes covered by the AMS is small. The AMS was operated at 1 Hz in FastMS mode (i.e., 25 s open, 5 s closed) and averaged to 30 s resolution for the data archive. The uncertainty of AMS observations measured during ACTIVATE is reported to be up to 50% based on processing assumptions
related to collection efficiency.

**Table 1: List of instruments and data sets used in this study, including their respective resolution, measurement type, and aircraft location.**

| Instrument | Variables | Resolution (temporal/vertical) | Measurement Type | Aircraft |
|---|---|---|---|---|
| DMT Cloud Condensation Nuclei (CCN) Counter | CCN concentration at given supersaturation (S) | 1 s | In Situ | HU-25 Falcon |
| High Spectral Resolution Lidar 2 (HSRL-2) | Aerosol backscatter coefficient (532 nm), Aerosol ID | 10 s /15 m | Remote Sensing | King Air |
| Scanning Mobility Particle Sizer (SMPS) | Aerosol size distribution (diameter = 0.003-0.1 µm) | 45 s | In Situ | HU-25 Falcon |
| Laser Aerosol Spectrometer (LAS) | Aerosol size distribution (diameter = 0.1-3.5 µm) | 1 s | In Situ | HU-25 Falcon |
| Aerodyne High Resolution Time-of-Flight Aerosol Mass Spectrometer (HR-ToF-AMS) | Non-refractory chemically resolved mass concentration | 30 s | In Situ | HU-25 Falcon |
| Diode Laser Hygrometer (DLH) | Ambient relative humidity (RH) | 1 s | In Situ | HU-25 Falcon |



## 3.2 κ-Köhler Theory

The activation of aerosols into cloud droplets is described by Köhler theory, in which the water vapor supersaturation in stable equilibrium with a condensed water droplet is a function of the particle radius. For a constant water vapor supersaturation, particles larger than a critical diameter will experience uncontrolled water condensation and growth to form a cloud droplet (Köhler, 1936). For calculations of $CCN_{theory}$ in this study, we use κ-Köhler theory, which uses a single, bulk hygroscopicity parameter kappa (κ) to represent the relative hygroscopicities of individual aerosol components (Petters & Kreidenweis, 2007). Using this methodology, the critical diameter ($D_{crit}$) of activation can be calculated with Eq. (1),

$$D_{crit} = \left(\frac{4A^3}{27\kappa ln^2 S_c}\right)^{1/3} ,$$
(1)

where $S_c$ is the specified instrument supersaturation during ACTIVATE, and A is defined in Eq. (2),

$$A = \frac{4\sigma_{s/a}M_w}{RT\rho_w} ,$$
(2)

where $\sigma_{s/a}$ is droplet surface tension, which is assumed to be a constant equivalent to that of pure water (0.0728 N m$^{-1}$; Petters & Kreidenweis, 2007), $M_w$ is the molecular weight of water (18.01528 g mol$^{-1}$), R is the universal gas constant (8.3145 J mol$^{-1}$ K$^{-1}$), T is temperature (298.15 K), and $\rho_w$ is the density of water (1000 kg m$^{-3}$).

### 3.2.1 Kappa Calculations from AMS Data

As shown in Eq. (1), $D_{crit}$ depends on a bulk kappa value representing aerosol chemical composition, and we calculate it using AMS observations and the Zdanovskii-Stokes-Robinson (ZSR) mixing rule (Zdanovskii, 1948; Stokes & Robinson, 1966) given in Eq. (3),

$$\kappa = \sum_i \varepsilon_i \kappa_i,$$
(3)

where $\varepsilon_i$ represents the volume fraction of each chemical component and $\kappa_i$ is the hygroscopicity value of each component. This set of calculations is done using the collocation-averaged AMS data associated with each collocated data point (Appendix A) and a histogram of calculated kappa values for all three aerosol types is given in Fig. 3. Additionally, we show a literature-average kappa range for each aerosol type, along with the standard deviation for each end of the range. These values are calculated using six literature values per aerosol type, including SFS (Carrico et al, 2008; Petters et al., 2009; Cerully et al., 2011; Engelhart et al., 2012; Bougiatioti et al., 2016; Gomez et al., 2018; Twohy et al., 2021), MPM (Andreae & Rosenfeld, 2008; Pringle et al., 2010; Gaston et al., 2018; Quinn et al., 2019; Miyazaki et al. 2020; Gong et al., 2023), and URB (Andreae & Rosenfeld, 2008; Pringle et al., 2010; Hung et al., 2014; Kim et al., 2017, Cai et al., 2018, Zamora et al., 2019). Note that Cerully et al. (2011) and Engelhart et al. (2012) provide the same range for SFS, but this value is only counted once in the average. Overall, calculations tend to agree well with those seen in the literature. Typical kappa values for marine and polluted



marine aerosols can vary widely depending on the amount of pollution in a region or if observations are made in cleaner, more remote areas.


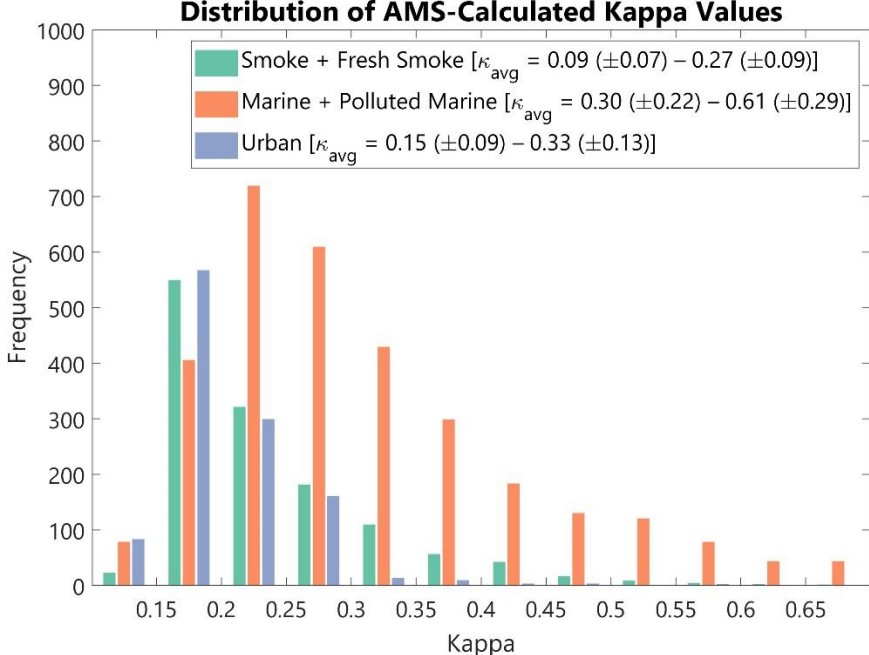

**Figure 3: Distribution of kappa values calculated using the methodology in Section 2.3.1 for each aerosol type. Literature average ranges, with standard deviation, for each aerosol type are given in parentheses.**

### 3.2.2 Critical Diameter ($D_{crit}$) and $CCN_{theory}$ calculation

After calculating kappa following the methodology in Sect. 3.2.1, we use Eq. (1) to calculate $D_{crit}$ and calculate $CCN_{theory}$ using Eq. (4),

$$CCN_{theory}\ (S) = \int_{D_{crit}}^{D_{max}} \frac{dN}{dlogD_p}\ dlogD_p, \tag{4}$$

where $D_{max}$ signifies the largest bin from the combined SMPS and LAS number size distribution (Schmale et al. 2018; Patel et al., 2021) and S is the CCN counter supersaturation. For direct comparisons between $CCN_{theory}$ and $CCN_{obs}$, we use the exact

CCN counter supersaturation value reported for each collocated data point within a small range of 0.36-0.38% (Sect. 4.1). For analyses using only $CCN_{theory}$ without a comparison to $CCN_{obs}$, we use a constant 0.37% supersaturation (Sect. 4.2-4.3).

### 3.3 Mie Calculations

The properties of light scattered by atmospheric aerosols are described by Mie theory, where aerosols are assumed to be

homogeneous, spherical, and have a diameter approximately equal to the wavelength of incident radiation (Mie, 1908). For



our calculations of BSC$_{theory}$, we calculate size-resolved particle backscattering efficiencies (Q$_{bsc}$) using the Mie scattering program by Bohren & Hoffman (1998) implemented in the libRadtran library of radiative transfer routines and programs (Emde et al., 2016). The two inputs needed to calculate Q$_{bsc}$ are particle size and a complex refractive index. We use typical refractive index values as retrieved by the Aerosol Robotic Network (AERONET) for different aerosol types to inform our

refractive index selection (Dubovik et al., 2002), with exact values given in Table B1.

For particle size input we use the SMPS and LAS size distribution bin diameters. However, here we must account for a significant difference in how in situ and HSRL-2 observations are made. With these BSC$_{theory}$ calculations, we want to model ambient BSC$_{obs}$ from the HSRL-2 to understand the relationship with in situ CCN$_{obs}$. However, since in situ instruments dry ambient air before collecting measurements, we need to account for the change in particle diameters due to water uptake at

ambient RH conditions, since particle size has a significant impact on the magnitude of light scattered. Calculations made to account for changes in particle diameter and refractive index due to hygroscopic growth are outlined in Appendix B. After these adjustments, humidified bin diameters (D$_{wet}$) and refractive index components (m$_{wet}$ and n$_{wet}$) are the final inputs into the Mie scattering calculations run in libRadtran. The size-resolved Q$_{bsc}$ values returned from these calculations are used to calculate BSC$_{theory}$ from the full aerosol size distribution, as shown in Eq. (5),

$$BSC_{theory} = \int_{r_1}^{r_n} \pi r^2 \, Q_{bsc} \, n(r) dr, \tag{5}$$

where n(r)dr represents the aerosol number concentration in each bin, and r$_n$ represents the largest bin in the SMPS and LAS combined size distribution. This set of calculations is done using the collocation-averaged size distribution data associated with each collocated data point.

**3.4 Data Filtering**

All input data for κ-Köhler and Mie calculations come from the collocated data set used for the observational analysis in Section 2. Each collocated data point contains an average value of CCN$_{obs}$, BSC$_{obs}$, as well as an average combined SMPS and LAS size distribution and set of AMS observations. Therefore, to enable a direct comparison between CCN$_{theory}$ and CCN$_{obs}$, as well as BSC$_{theory}$ and BSC$_{obs}$, this collocated data set is used throughout the entirety of the study. In this section we describe

several filtering steps that are performed to minimize potential errors in the subsequent analyses. Some are motivated by the observational methodology taken in L23, and others are specific to the CCN$_{theory}$ and BSC$_{theory}$ calculations. All steps are summarized in Fig. 4.

We begin with the filtering criteria applied to data in the CCN$_{obs}$ − BSC$_{obs}$ relationships shown in Section 2. Since these data points are the basis for the rest of the analysis, each of these filtering steps also applies to data used for calculating

CCN$_{theory}$ and BSC$_{theory}$. As we are analyzing these relationships by aerosol type, we start by removing observations from the collocated data set where an HSRL-2 Aerosol ID is not determined, which typically occurs if the full set of HSRL-2 observables are not available. This step removed 51% of the collocated data set. Additionally, we remove any points where the collocation method averages across varying Aerosol IDs to avoid introducing additional uncertainty into the aerosol type. Similarly, we





remove points where the standard deviation is greater than the mean of CCN concentration that fall within our collocation

criteria to avoid potential errors due to large variability or gradients in aerosol concentration. Lastly, we remove any collocated

points where fewer than two samples comprise the average. This is done to reduce potential noise in the data set, especially

for the in situ size distribution data that have a critical role in both sets of theoretical calculations. In general, collocated data

points represent an average of 10 observations from the 1-second merged in situ data files, some of which have a lower original

resolution (Table 1). Each of these filtering steps is applied to all data in this study, and the impact of each step on the total

number of collocated data points is shown in Fig. 4.

In L23, we found that the correlation between CCN concentration and HSRL-2 observations was strongest for

supersaturations greater than 0.25%. Additionally, since CCN strongly depends on supersaturation, we limit observations to a

small range of supersaturation values to reduce additional variability. Therefore, for analyses that are strictly observational

(Sect. 2) or that compare theoretical calculations with observations (Sect. 4.1), we limit our collocated data set to a

supersaturation range of 0.36-0.38%. This range was chosen due to a supersaturation of 0.37% being the most frequent value

during ACTIVATE. This step is only applied to analyses that include observations because for calculations of $CCN_{theory}$ we

apply a constant supersaturation of 0.37% to any observed size distribution. That is, we do not unnecessarily limit the data

used for theoretical calculations by filtering according to CCN counter supersaturation.

The last data filtering step serves as a check to $CCN_{theory}$ calculations. In addition to using Eq. 1 to calculate $D_{crit}$, as

described in Sect. 3.2, we also use an estimation method to validate κ-Köhler calculated values. This method integrates the

combined SMPS and LAS number size distribution from largest toward smallest bin diameters until the difference between

the summed aerosol concentration and observed CCN concentration reaches a minimum. We refer to the bin diameter where

this difference reaches a minimum as the estimated $D_{crit}$ ($D_{crit,est}$). We compare these values to the κ-Köhler calculated values

($D_{crit,calc}$) and require that $D_{crit,calc}$ values fall within ±20% of the $D_{crit,est}$ values. This step ensures that our calculated $CCN_{theory}$

values will closely match $CCN_{obs}$ values and removes size distributions that may have higher noise or several bins with missing

concentrations. The threshold of ±20% is chosen to correspond to the SMPS and LAS reported uncertainty that impacts the

accuracy of the $D_{crit,est}$ value. This step applies to all analyses involving calculations of $CCN_{theory}$ and $BSC_{theory}$ (Sect. 4).

As seen in Fig. 4, some of these steps remove a significant amount of data from the analysis. While the amount of

data removed was taken into consideration at each step, all steps were taken as a precaution against introducing unnecessary

variation and uncertainty into the analysis. The application of slightly different combinations of filtering steps to the analyses

in Sect. 4 was done intentionally to allow for as much data as possible to be included in each step. Therefore, while the $D_{crit}$

agreement filtering step is applied everywhere we calculate $CCN_{theory}$ and $BSC_{theory}$, the CCN counter supersaturation filter is

only applied where it needs to be used to control the supersaturation dependence of $CCN_{obs}$. Since the goal of this study is to

understand the relationship between $CCN_{obs} - BSC_{obs}$ through the lens of the theoretical calculations, removal of extraneous

noise and variability from the input data allows for analyses to more accurately determine the true underlying factors governing

the $CCN_{theory} - BSC_{theory}$ relationship. We discuss a comparison between observed and theoretically calculated CCN and BSC

in Sect. 4.1, but a detailed discussion of closure for these variables is beyond the scope of this study.





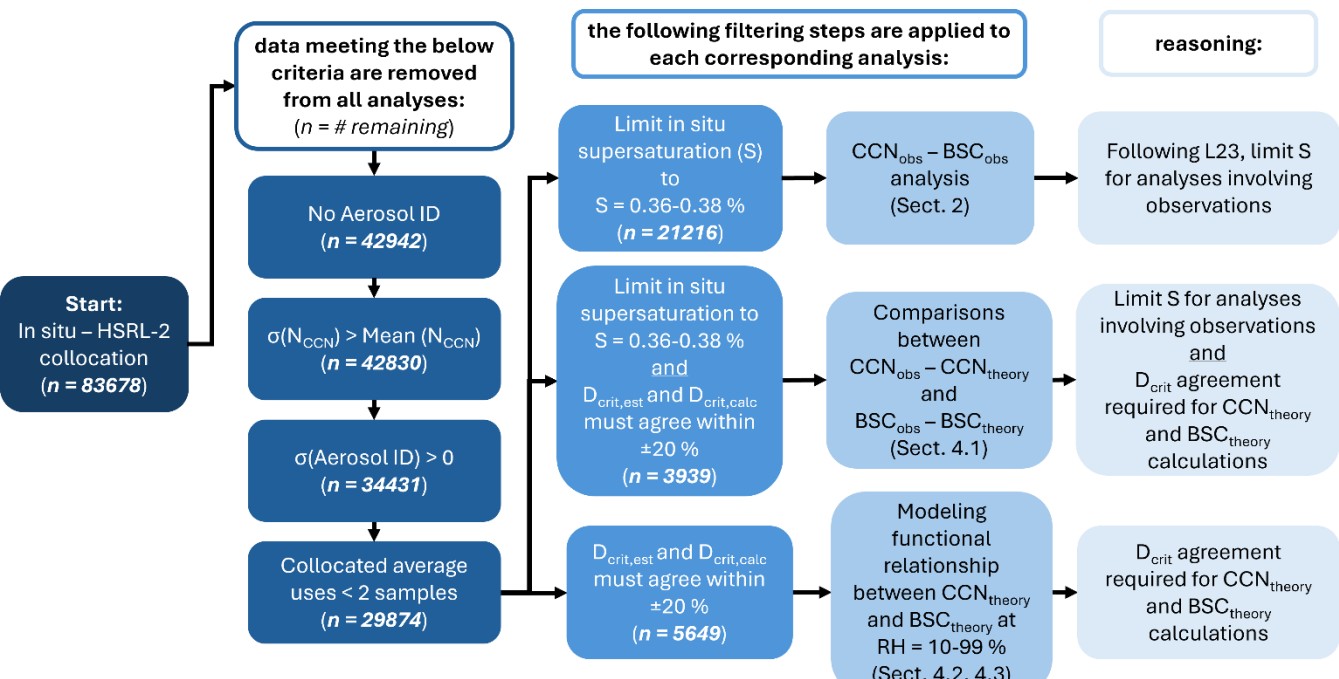

**Figure 4: Flowchart describing the data filtering steps applied to data for all analyses and filtering steps applied to data for specific analyses. The number of points remaining after each step (n) is given in parentheses.**

## 4 Results

### 4.1 CCN and BSC Observations vs. Calculations

We first calculate $CCN_{theory}$ and $BSC_{theory}$ under observed ambient conditions and compare calculations to observations. $CCN_{theory}$ is calculated for all data using the corresponding CCN counter supersaturation, and $BSC_{theory}$ is calculated from humidified aerosol size distributions using the corresponding observed RH value. Since this step involves theoretical calculations and comparison with observations, we limit the data set to observations made at CCN counter supersaturation between 0.36-0.38% and apply the $D_{crit}$ agreement filtering step (Fig. 4).

The comparison between $CCN_{obs}$ and $CCN_{theory}$ is given in Fig. 5. We show results requiring a $D_{crit}$ agreement outlined in gray, while calculations without this requirement are plotted in the background. Results for calculations not requiring $D_{crit}$ agreement are shown to demonstrate how this requirement impacts the data set. Results of a linear regression between $CCN_{obs}$ and $CCN_{theory}$ for data requiring the $D_{crit}$ agreement show that for all aerosol types $R^2$ ranges from 0.91-0.94 and RMSE ranges from 87-133 cm$^{-3}$. These RMSE values are very close to the approximate median value of CCN uncertainty of 150 cm$^{-3}$. Data are generally clustered very close to the 1:1 line for all aerosol types, and the lines of best fit also fall close to the 1:1 line.



Overall, this analysis gives us confidence that our methodology accurately calculates CCN$_{theory}$, as a necessary precursor for the correlation analysis with BSC$_{theory}$.

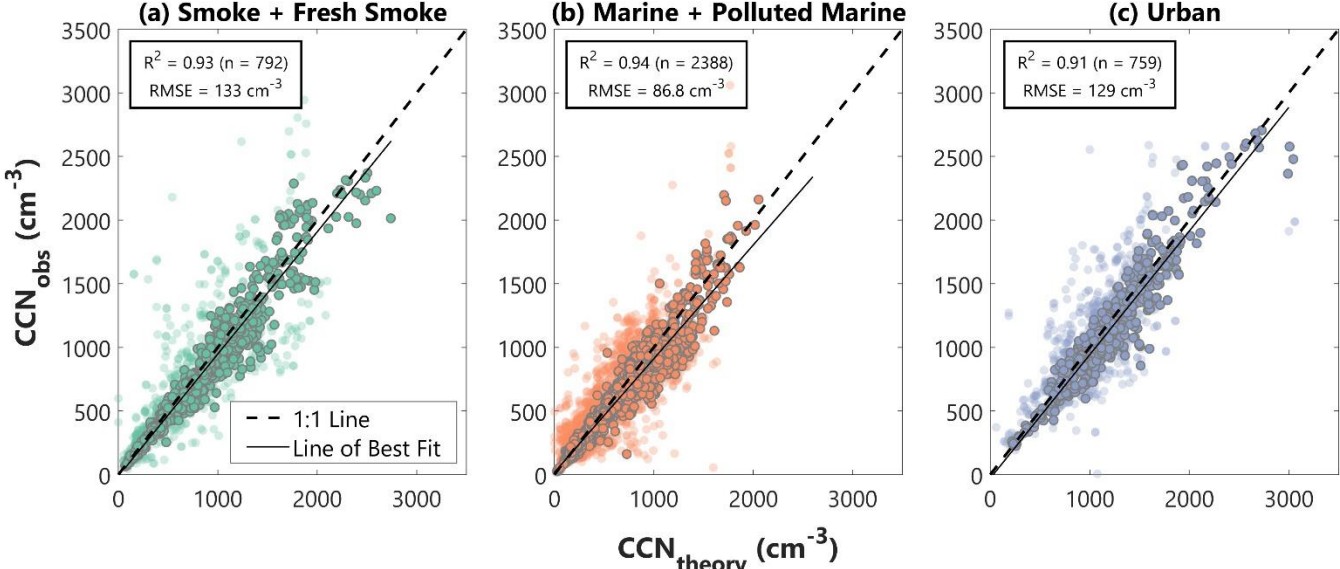

**Figure 5: CCN$_{obs}$ vs. CCN$_{theory}$ for (a) smoke and fresh smoke, (b) marine and polluted marine, and (c) urban aerosols. The 1:1 lines are dashed, and the lines of best fit for the linear regressions between both variables are solid.**

The same statistics are shown for our comparison between BSC$_{obs}$ and BSC$_{theory}$ in Fig. 6 where marker colors correspond to RH to show the impact of hygroscopic growth on calculated BSC$_{theory}$. Here, our $R^2$ values range from 0.45-0.75, and RMSE ranges from 2.7E-04 to 1.6E-03 km$^{-1}$sr$^{-1}$. We find that the performance of our calculations does not appear to systematically decline for observations made at high RH, providing confidence in our humidification calculation methods. The $R^2$ and RMSE values indicate a weaker correlation between observations and calculations than for CCN, but data remain primarily clustered around the 1:1 line. While use of the D$_{crit}$ filtering step for this analysis and subsequent removal of size distributions with higher noise or missing concentrations does benefit BSC$_{theory}$ calculations, it does not force a degree of agreement between BSC$_{obs}$ and BSC$_{theory}$ in the same way that it does for agreement between CCN$_{obs}$ and CCN$_{theory}$. Additionally, CCN$_{obs}$ and the inputs for the CCN$_{theory}$ calculation all come from in situ observations, while the BSC$_{theory}$ calculation uses in situ observations as input but is compared to BSC$_{obs}$ from remote sensing instrumentation on a separate platform. Varying resolutions and collocation averaging between in situ and HSRL-2 observations may cause discrepancies between BSC$_{obs}$ and BSC$_{theory}$. As with the CCN comparison (Fig. 5), this analysis also gives us confidence that our methodology results in BSC$_{theory}$ values of a similar magnitude as BSC$_{obs}$.





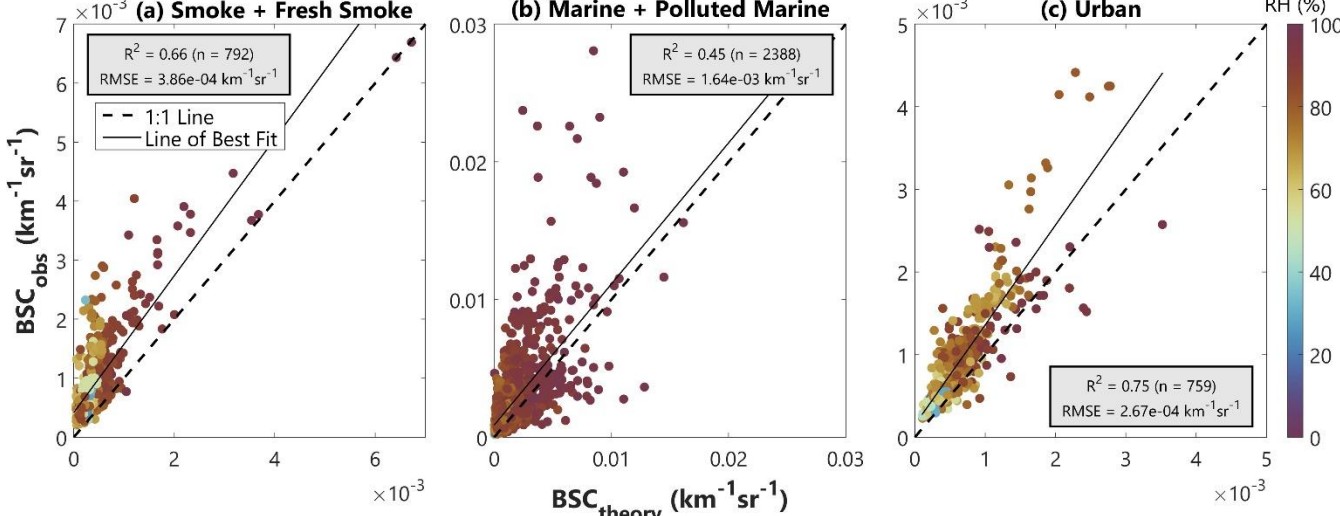

Figure 6: BSC$_{obs}$ vs. BSC$_{theory}$ for (a) smoke and fresh smoke, (b) marine and polluted marine, and (c) urban aerosols. The 1:1 lines are dashed, and the lines of best fit for the linear regressions between both variables are solid. Marker colors correspond to ambient RH that was observed by each BSC$_{obs}$ and applied to calculate each corresponding BSC$_{theory}$.

## 4.2 Estimating Predictor Importance

In investigating the CCN$_{obs}$ – BSC$_{obs}$ relationship for different aerosol types, we determined that a linear regression is not an appropriate model for the ACTIVATE data (Fig. 2). Additionally, we have shown reasonable agreement between CCN$_{obs}$ and CCN$_{theory}$ and between BSC$_{obs}$ and BSC$_{theory}$ at ambient conditions. Next, we use the theoretical calculations to investigate and interpret causes of scatter and non-linearity in the CCN$_{obs}$ – BSC$_{obs}$ relationship. Analyses in this and the next section use CCN$_{theory}$ calculated at a constant supersaturation of 0.37%.

Recall that the three main factors influencing how CCN concentration relates to AOPs are ambient RH, the shape of the aerosol size distribution, and aerosol chemical composition. Due to the highly interconnected nature of these factors and their relationships with CCN and BSC, we use random forest (RF) models to determine the relative importance of each factor in controlling the CCN$_{theory}$ – BSC$_{theory}$ relationship. A random forest is an ensemble of decision trees where each tree is created using the best split from a randomly selected subset of predictors. The final prediction comes from a majority vote among individual trees (Breiman, 2001; Hu et al., 2017). This method was chosen due to its high accuracy, generalization capability, ability to handle non-linear relationships between features, and ability to provide estimates of predictor importance. Another benefit of this method is the ability to consider all input variables collectively, as opposed to investigating or perturbing individual input variables one at a time. For each model, we use 200 ensemble learning cycles and specify that all predictor variables are used at each node to ensure that each tree uses all predictor variables. Ten-fold cross-validation is used during training to prevent overfitting by any single model. The final predictor importance is determined by averaging the importance



estimates across the 10 models, and the standard deviation is used to reflect the variations in the final calculated predictor importance estimates. Additionally, we do not separate our data into training and testing subsets, because our purpose is not to train and refine a model that predicts $CCN_{theory}$ or the $CCN_{theory} - BSC_{theory}$ relationship. Redemann & Gao (2024) provide a well-tested machine learning method with which CCN concentration is predicted from several HSRL-2 and reanalysis input variables. Rather, here we use RF predictor importance as a tool to help investigate the impact that ambient RH, aerosol size distribution, and aerosol chemical composition each have on the $CCN_{theory} - BSC_{theory}$ relationship.

We use a combination of observed effective radius ($R_{eff}$), geometric mean radius (GMR), RH, and kappa as predictors of the $CCN_{theory}:BSC_{theory}$ ratio in our RF models. Effective radius is the ratio of the $3^{rd}$ and $2^{nd}$ moments of the aerosol size distribution, sometimes called the area-weighted mean radius. This makes it useful for optical measurements as the energy removed from light by an aerosol is proportional to its area. Effective radius is calculated using Eq. (6),

$$R_{eff} = \frac{\int_0^\infty \pi r^3 n(r) dr}{\int_0^\infty \pi r^2 n(r) dr},$$ (6)

where r is particle radius and n(r)dr is the aerosol concentration within each bin of the size distribution. Geometric mean radius is the mean of the aerosol size distribution in log space, as given by Eq. (7),

$$GMR = \left( \frac{\int_0^\infty \ln r \, n(r) dr}{N_0} \right),$$ (7)

where $N_0$ is the total number of particles in the size distribution. While both $R_{eff}$ and GRM capture the shape of the size distribution, we use both parameters because of their different magnitudes and varying information content. The weighting of $R_{eff}$ toward larger particles increases its relevance for AOPs, while GMR tends to fall within the fine mode of the size distribution closer to $D_{crit}$ and aerosol sizes relevant for CCN activation. We train the RF models to predict the ratio of $CCN_{theory}:BSC_{theory}$ based on this combination of input variables.

First, we train a model for all aerosol types combined. Here, the Aerosol ID from our collocated in situ and remote sensing data set is added as an additional predictor to test the dependence of $CCN_{theory}:BSC_{theory}$ on lidar-indicated aerosol type. Average relative predictor importance estimates across all 10 folds are shown in Fig. 7a with a standard deviation designated for each average. Overall, $R_{eff}$ is determined to be the most important predictor of $CCN_{theory}:BSC_{theory}$, followed by RH. Aerosol ID is the $3^{rd}$ most important predictor, and GMR and kappa are approximately equal as the $4^{th}$ and $5^{th}$ most important predictors, respectively.

Next, we train three individual models that predict $CCN_{theory}:BSC_{theory}$ for each individual aerosol type as separated by Aerosol ID and again average the relative predictor importance estimates across all 10 folds (Fig. 7b). We find that after separating aerosol types, $R_{eff}$ remains the most important predictor of $CCN_{theory}:BSC_{theory}$ for all aerosol types. Kappa ranking least important for each aerosol type indicates that separating aerosol types using the Aerosol ID adequately constrains the impact of aerosol chemical composition on the $CCN_{theory} - BSC_{theory}$ relationship. These separate models also indicate that RH is the second most important predictor for all aerosol types.





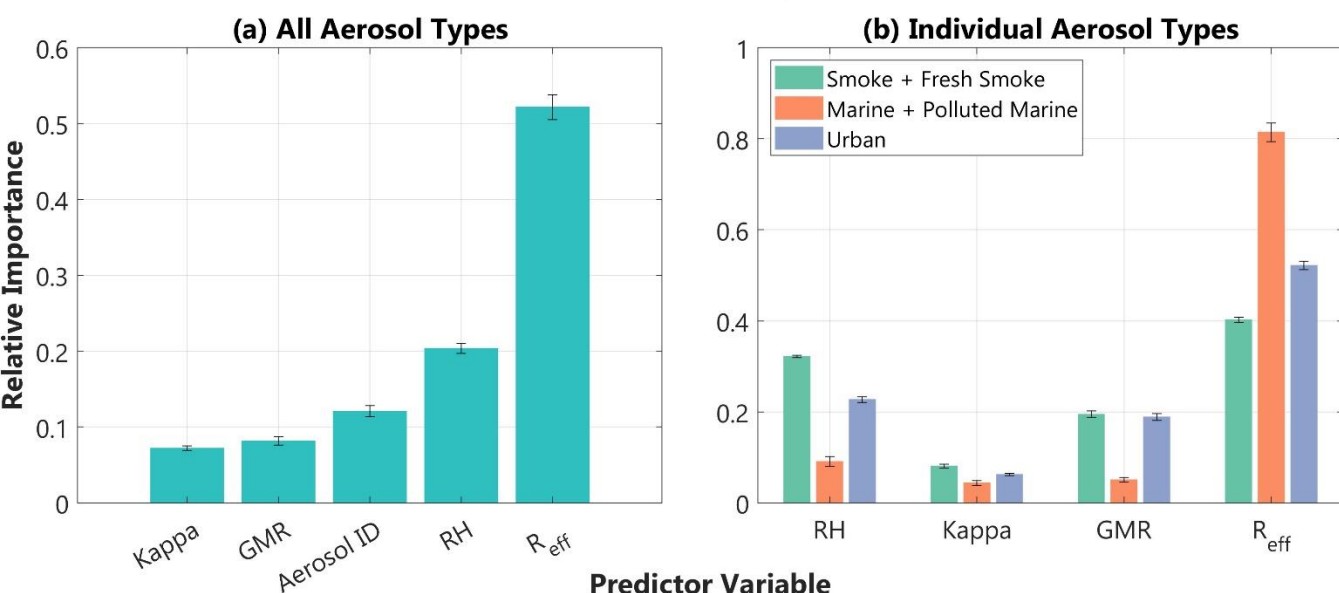

**Figure 7: Average random forest predictor importance estimates across 10-fold cross-validation for (a) the model run for a combination of aerosol types combined and (b) individual models run for the three different aerosol types. Each model predicted the $CCN_{theory}$:$BSC_{theory}$ ratio based on the observed input variables listed on each x-axis. All importance estimates are relative. Error bars designate standard deviation across the 10-fold cross-validation.**

### 4.3 Modeling $CCN_{theory}$:$BSC_{theory}$ using Effective Radius

Based on the RF predictor importance estimate indication that $R_{eff}$ is the most important predictor for the $CCN_{theory} - BSC_{theory}$ relationship compared to RH, kappa, and GMR (Fig. 7), we now investigate the physical relationship between $R_{eff}$ and the $CCN_{theory}$:$BSC_{theory}$ ratio. We focus on $R_{eff}$ to further explore and understand the RF indication of its high importance compared to the other predictors and to understand how much variance in $CCN_{theory}$:$BSC_{theory}$ can be explained by $R_{eff}$ alone.

We start by humidifying each dry aerosol size distribution at 10% RH increments from 10-99% and calculating $CCN_{theory}$, $BSC_{theory}$, and $R_{eff}$ from each humidified size distribution. This process allows us to model all variables at a wide range of plausible environmental RH values that are not constrained to observed ambient conditions and to form a more comprehensive understanding of the underlying physical relationship between $R_{eff}$ and $CCN_{theory}$:$BSC_{theory}$. When comparing $CCN_{theory}$:$BSC_{theory}$ and $R_{eff}$, we fit two-term exponential curves for each aerosol type to represent the relationship (Fig. 8). A two-term exponential was chosen for each aerosol type due to a slightly higher $R^2$, lower RMSE, and better visual fit to the





larger $R_{eff}$ values than a one-term exponential fit. For each aerosol type we provide the $R^2$ and RMSE (Fig. 8), and fit coefficients are provided in Table 2.

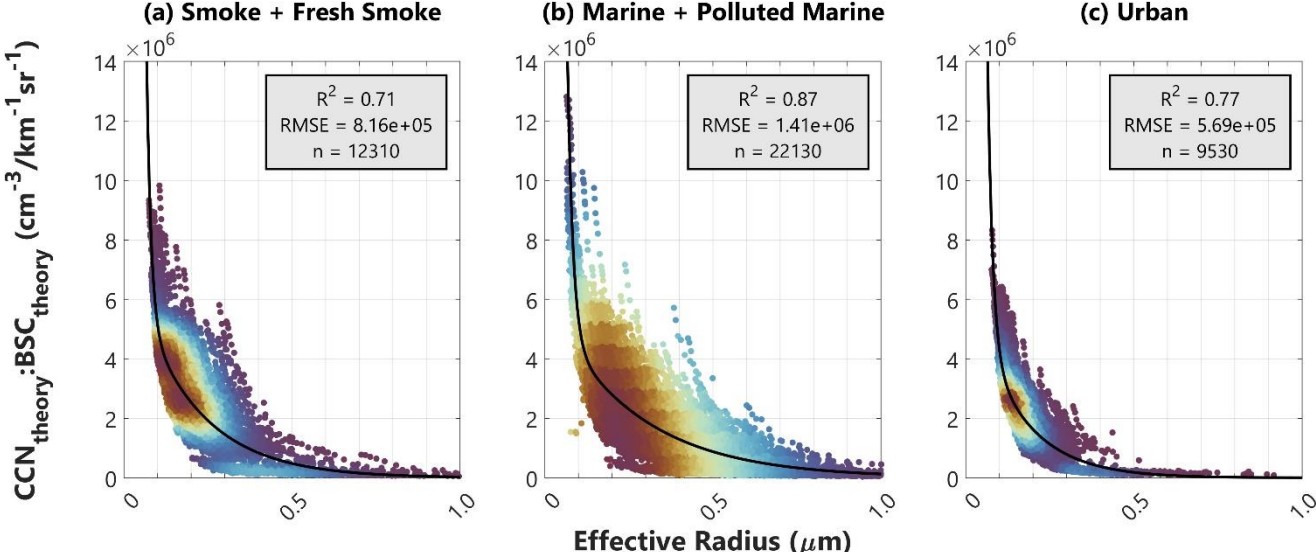

**Figure 8: CCN$_{theory}$:BSC$_{theory}$ vs. R$_{eff}$ for (a) smoke and fresh smoke, (b) marine and polluted marine, and (c) urban aerosols. Marker colors correspond to the density of surrounding points, with red shades indicating high density and blue shades indicating lower density. The black line represents the two-term exponential curve fit for each aerosol type. R$^2$, RMSE, and number of data points (n) for each exponential fit are also provided.**

**Table 2: Coefficients for the two-term exponential curves fit to each aerosol type to model the CCN$_{theory}$:BSC$_{theory}$ − R$_{eff}$ relationship. All fit equations take the form of $y = a_1 exp(b_1 x) + a_2 exp(b_2 x)$, where y corresponds to the CCN$_{theory}$:BSC$_{theory}$ ratio in cm$^{-3}$/km$^{-1}$sr$^{-1}$ and x corresponds to R$_{eff}$ in µm. Relative uncertainties for each coefficient are given in parentheses, estimated using 95% confidence bounds.**

| | $a_1$ (cm$^{-3}$/km$^{-1}$sr$^{-1}$) | $b_1$ (µm$^{-1}$) | $a_2$ (cm$^{-3}$/km$^{-1}$sr$^{-1}$) | $b_2$ (µm$^{-1}$) |
|---|---|---|---|---|
| **Smoke + Fresh Smoke** | 1.417E09 (26.4%) | -75.64 (4.2%) | 7.627E06 (1.4%) | -5.512 (1.4%) |
| **Marine + Polluted Marine** | 6.134E08 (1.3%) | -62.01 (0.6%) | 6.028E06 (1.1%) | -3.846 (1.4%) |
| **Urban** | 7.032E08 (21.3%) | -65.73 (3.9%) | 6.919E06 (2.4%) | -7.231 (1.9%) |

Next, we use each of these two-term exponential fits to calculate CCN:BSC from values of $R_{eff}$ in our ambient collocated data set and compare to CCN$_{theory}$:BSC$_{theory}$. Here, we refer to CCN:BSC modelled using the two-term $R_{eff}$ exponential fits as "(CCN:BSC)$_{model}$" to capture that the ratio itself is modelled using $R_{eff}$, not each term individually, and to



distinguish it from $CCN_{theory}$:$BSC_{theory}$. This comparison is shown in Fig. 9, where we find that overall, most data is clustered around the 1:1 line for each aerosol type. We see that RMSE and mean relative error (MRE) are lowest for the URB category

and highest for MPM. Additionally, SFS and URB have many data points at or slightly below the 1:1 line and a majority of $(CCN:BSC)_{model}$ ratios have magnitudes of about 2E06 to 4E06 $cm^{-3}/km^{-1}sr^{-1}$, while most values for MPM are less than 2E06 $cm^{-3}/km^{-1}sr^{-1}$. The $R^2$ values for all aerosol types range between 0.68-0.79.

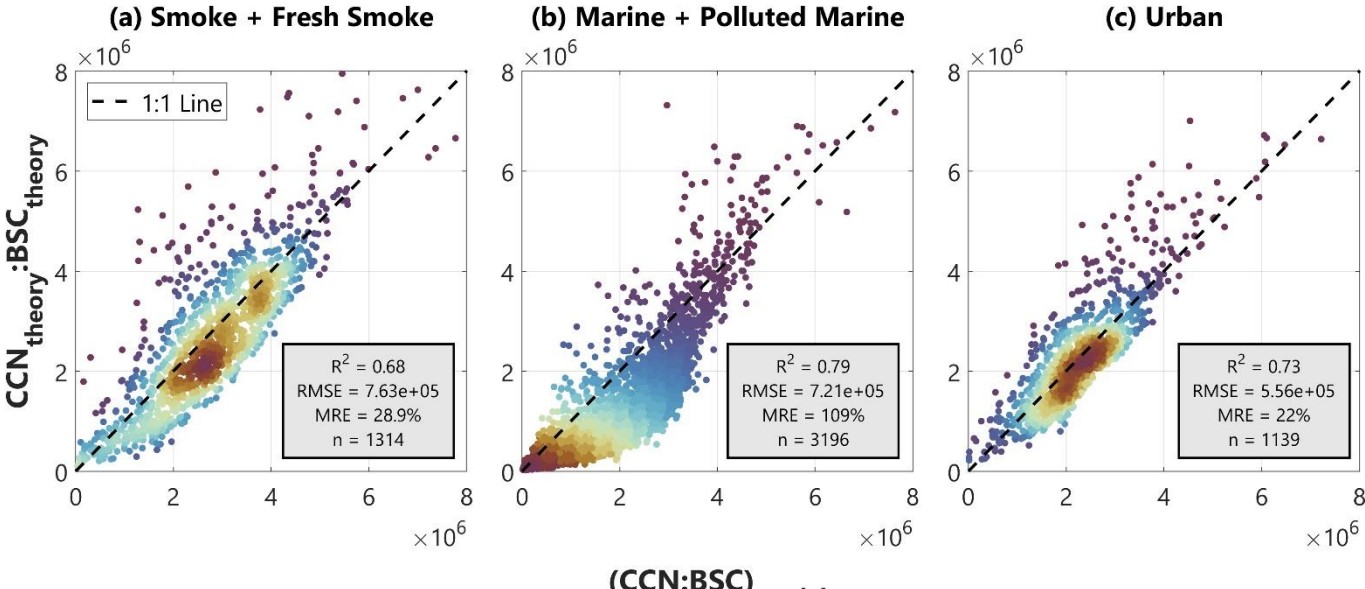

**Figure 9: Comparison of $CCN_{theory}$:$BSC_{theory}$ to $(CCN:BSC)_{model}$ for (a) smoke and fresh smoke, (b) marine and polluted**

**marine, and (c) urban aerosols. $(CCN:BSC)_{model}$ values come from the two-term exponential fits shown in Fig. 8 and defined in Table 2. The units for both axes are $cm^{-3}/km^{-1}sr^{-1}$. Marker colors correspond to the density of surrounding points, with red shades indicating high density and blue shades indicating lower density. The dashed line on each panel is the 1:1 line. $R^2$, RMSE, mean relative error (MRE), and number of data points (n) for each exponential fit are also provided.**

## 5 Discussion

### 5.1 Physical Relationships

Based on a set of predictors for the $CCN_{theory}$:$BSC_{theory}$ relationship including $R_{eff}$, GMR, kappa, and RH, RF predictor importance estimates indicated that $R_{eff}$ was the most important predictor for all three aerosol types of interest in this study. Therefore, we further investigated the relationship between $CCN_{theory}$:$BSC_{theory}$ and $R_{eff}$ for a wide range of plausible

environmental RH conditions and found a two-term exponential relationship. In further understanding this pattern, it is important to recall that $R_{eff}$ is influenced more significantly by coarse mode than fine mode particles. As the coarse mode



number concentration increases, we expect $BSC_{theory}$ to increase more compared to $CCN_{theory}$, thus decreasing the $CCN_{theory}:BSC_{theory}$ ratio. This finding is similar that of Shen et al. (2019), where an exponential relationship was found between a CCN:AOP ratio and the geometric mean diameter of generated lognormal unimodal size distributions. Additionally, the

exponential fits here show a steeper decrease in $CCN_{theory}:BSC_{theory}$ for MPM aerosols compared to other aerosol types (Table 2). Since MPM aerosols are expected to have a more significant coarse mode contribution, it appears that the effect of $BSCt_{heory}$ increasing more than $CCN_{theory}$ is more pronounced for this aerosol type.

Based on the $R^2$ values of 0.68-0.79 in our comparison of $CCN_{theory}:BSC_{theory}$ and $(CCN:BSC)_{model}$ (Fig. 9), we find that modelling $CCN_{theory}:BSC_{theory}$ using two-term exponential $R_{eff}$ relationships can explain approximately 68-79% of the

variance in the $CCN_{theory} - BSC_{theory}$ relationship. Whereas we previously hypothesized in L23 that aerosol hygroscopic growth at high ambient RH may be the leading cause of variability when relating $CCN_{obs}$ and $BSC_{obs}$, here we find that most variability is attributable to differences in $R_{eff}$. Furthermore, this analysis also speaks to inherent differences in how $CCN_{theory}$ relates to $BSC_{theory}$ for different aerosol types. We find that when using a large set of actually observed aerosol size distributions as the input to theoretical calculations, there is a significant difference in the range of possible $R_{eff}$ values for each aerosol type (Fig.

8). For example, many MPM observations span a wide range of $R_{eff}$ between approximately 0.1-0.5 μm while SFS and URB, even at wide range of possible ambient RH, primarily see $R_{eff}$ values limited to a small range between 0.1-0.2 μm. Additionally, when we look at the magnitude of $CCN_{theory}:BSC_{theory}$ values for each aerosol type, MPM tends to have much lower values than SFS and URB (Fig. 9). Higher $R_{eff}$ values in addition to a higher likelihood for hygroscopic growth in humid marine environments act to increase $BSC_{theory}$ more than $CCN_{theory}$, thus decreasing the $CCN_{theory}:BSC_{theory}$ ratio more than for other

aerosol types.

Here we present three $CCN_{theory}:BSC_{theory} - R_{eff}$ exponential fits as a methodology for explaining variance in the $CCN_{theory} - BSC_{theory}$ relationship. The exact functional forms presented in Fig. 8 are most appropriate for ACTIVATE observations, and the coefficients would likely need to be adjusted before applying to other data sets. While we expect the general exponential pattern to hold for other data sets, any differences in observed aerosol size distribution or chemical

composition would likely change the exact fit coefficients.

**5.2 Implications for Remote Sensing Techniques**

This study indicates several important considerations for future work constraining $N_{CCN}$ from remote sensing observations and future spaceborne lidar data sets. Most importantly, given our finding that particle size, as parameterized by $R_{eff}$, is the most

important predictor in determining the $CCN_{theory} - BSC_{theory}$ relationship, this suggests two key points. First, a simple linear approximation with $BSC_{obs}$ will not well-constrain $CCN_{obs}$ in most cases. Many previous studies have suggested that the relationship between CCN concentration and various AOPs is often non-linear, specifically for AOD. Considering this background, the results here suggest that variations in aerosol size distribution may be a leading cause of non-linearity when using AOD as a proxy for CCN concentration. Seemingly in contrast with the results presented here, in L23 we investigated

the relationship between CCN concentration and aerosol index (AI), an indicator of particle size, and found little to no



difference between CCN – AI and the CCN – EXT or CCN – BSC relationships. Therefore, for observations of smoke at low RH over the ORACLES region, we concluded that there was a very small variation of aerosol size in the observations. With minimal differences in aerosol size and most smoke plume observations made at low ambient RH, conditions permitted a simple linear approximation to relate $CCN_{obs}$ and $BSC_{obs}$. On the contrary, the larger data set from the ACTIVATE campaign

is characterized not only by a variety of aerosol types, but by a wider range of aerosol size distributions and a higher fraction of observations made at high ambient RH in the MBL, all of which contribute to increased non-linearity between $CCN_{obs}$ and $BSC_{obs}$.

Related to this non-linearity, a second key point from this analysis is that in most cases, efforts to constrain CCN concentration using AOPs need to include a measure of the aerosol size distribution to accurately represent variability in the

relationship. Here, we have taken advantage of the availability of in situ aerosol size distributions and represented them using $R_{eff}$. However, to constrain CCN concentration solely from spaceborne lidar observations, our findings suggest that either satellite retrievals of $R_{eff}$ would need to be collocated with lidar observations, or a different lidar-derived indicator of aerosol size would need to be used. For example, AI can be calculated using two wavelengths of aerosol extinction from lidar, and other multi-wavelength parameters such as the lidar ratio or backscatter color ratio contain information about aerosol size that

could be tested in place of $R_{eff}$ for future methods based solely on a spaceborne lidar system. Additionally, $R_{eff}$ retrievals from the recently launched SPEXone multi-angle polarimeter onboard the NASA Plankton, Aerosol, Cloud, and ocean Ecosystem (PACE) mission (Hasekamp et al., 2019) is another option for quantifying aerosol size in CCN concentration estimates.

Lastly, when predicting $CCN_{theory}$:$BSC_{theory}$ for all aerosol types combined, the RF predictor importance estimates indicated that aerosol type, as represented by the HSRL-2 Aerosol ID, is the third most important predictor (Fig. 7a). Since the

Aerosol ID product categorizes aerosol types based on HSRL-2 optical properties, such as BSC, this may explain why Aerosol ID is estimated to be a more important predictor of $CCN_{theory}$:$BSC_{theory}$ than kappa in terms of aerosol type and chemical composition. This finding, in addition to the qualitative differences seen in the impact of high RH between aerosol types (Fig. 4), suggests that while Aerosol ID is not the most important predictor, separately analyzing the CCN – BSC relationship for different aerosol types provides insight into physical differences in CCN – AOP relationships between aerosol types.


### 5.3 Sources of Uncertainty and Limitations

There are several assumptions underlying both κ-Köhler and Mie theories in addition to uncertainties associated with the observations used as input. Individual instrument uncertainties are discussed in Sect. 2.1, and calculation assumptions are discussed in Sect. 2.3 and 2.4. Here we acknowledge the primary sources of uncertainty underlying this analysis and the

limitations to its applicability.

First, the most significant sources of uncertainty come from uncertainty associated with in situ observations. For example, we use AMS observations to calculate a bulk kappa value needed for κ-Köhler calculations. While we find that our calculated values are generally close to those found in the literature for all three aerosol types (Fig. 2), there are a variety of factors that may cause discrepancies. For example, the fraction of mass observed at sizes close to $D_{crit}$ is generally small,



meaning that AMS sensitivity to chemical composition at relevant CCN sizes can be limited. Additionally, κ-Köhler theory assumes that chemical composition is fixed across all aerosol sizes (Petters & Kreidenweis, 2007), which may cause discrepancies between $CCN_{obs}$ and $CCN_{theory}$. Additionally, Kim et al. (2017) found that CCN closure using AMS-calculated kappa values was less accurate than when using kappa calculated from humidified tandem differential mobility analyzer (HTDMA) observations.

We also consider observational uncertainty associated with in situ size distributions that impact both $CCN_{theory}$ and $BSC_{theory}$ calculations. For example, when considering the comparison between $BSC_{obs}$ and $BSC_{theory}$, we see the lowest $R^2$ for the MPM comparison (Fig. 6b), for which we present two possible causes. First, marine aerosols have a greater tendency compared to smoke and urban aerosols to be non-spherical in shape, as has been discussed for the ACTIVATE dataset by Ferrare et al. (2023), while Mie theory assumes that particles are spherical (von Hoyningen-Huene & Posse, 1996; Bi et al.

2018). Second, in situ aerosol size distributions tend to underrepresent coarse mode aerosols due to inefficient sampling at large sizes (McMurry, 1999; Ryder et al., 2018; Kangasluoma et al., 2020). Since marine aerosols tend to have a dominant coarse mode that contributes significantly to light scattering, and since this coarse mode is likely underrepresented by the in situ size distributions used as input to Mie calculations, this may be another cause of the discrepancy between $BSC_{obs}$ and $BSC_{theory}$. However, overall Fig. 5 and 6 provide confidence that the combination of uncertainties in the size distributions and

other input variables do not prohibit reasonable agreement between $CCN_{obs}$ and $CCN_{theory}$ or between $BSC_{obs}$ and $BSC_{theory}$. Therefore, while uncertainties in the in situ data are likely to cause errors in our theoretical calculations, the intermediate comparison between observations and calculations provides confidence that these uncertainties do not undermine the validity of this study.

            Lastly, there are a few important considerations for the applicability and limitations of this study. While the

ACTIVATE campaign collected many observations, our findings are limited to the campaign study area and have not been tested on other datasets. As previously mentioned, we would expect the general exponential relationship between $CCN_{theory}:BSC_{theory} - R_{eff}$ to hold for other data sets, but the exact fit coefficients would likely need to be adjusted.

**6 Conclusion**

To improve our understanding of CCN distributions, many techniques have developed proxies and parameterizations using

remotely sensed AOPs. Such strategies often provide a good constraint for CCN, but challenges remain due to factors such as aerosol hygroscopic growth and variations in the aerosol size distribution. In this study, we investigate the dominant governing factors of the $CCN_{theory} - BSC_{theory}$ relationship for different aerosol types using observation-informed theoretical calculations and find that $R_{eff}$ is the most important predictor for smoke, marine, and urban aerosols.

            This dependence of $CCN_{theory} - BSC_{theory}$ on the aerosol size distribution explains why, as expected, a linear

approximation generally is not an appropriate method for well-representing the relationship. Rather, this approach only works in limited, specific cases. For example, when analyzing the $CCN_{obs} - BSC_{obs}$ relationship for observations of smoke at low



ambient RH with a narrow range of aerosol sizes in ORACLES, a linear regression performed well. However, in cases such
as ACTIVATE where (i) most observations are made at high ambient RH, (ii) there are a variety of aerosol types present, and
(iii) there exists a wider range of observed aerosol size distributions, this approach is not possible. Through these observation-
informed analyses, we have provided a theoretical framework for understanding the impact of different governing factors on
the $CCN_{obs} - BSC_{obs}$ relationship and the relative importance of the size distribution compared to chemical composition and
hygroscopic growth at high ambient RH.

Our findings suggest a few key takeaways for future studies using spaceborne remote sensing instrumentation, such
as CALIPSO (Cloud-Aerosol Lidar and Infrared Pathfinder Satellite Observation) or other future spaceborne lidar
observations, to retrieve CCN concentrations at cloud-relevant altitudes. Most importantly, we found through using a wide
range of in situ observed size distributions that $R_{eff}$ well-captures the strong dependence of the $CCN_{theory} - BSC_{theory}$ relationship
on the aerosol size distribution. That is, for areas with a wide variety of observed size distributions, CCN cannot be well-
estimated from BSC without including aerosol size. Therefore, future remote sensing methods based on estimating $N_{CCN}$ from
particulate backscatter would require a lidar capable of providing $R_{eff}$, a backscatter lidar in combination with a polarimeter,
or collocated satellite retrievals of $R_{eff}$. Overall, we found that there is great benefit in using a wide variety of in situ observed
aerosol size distributions as an input for $CCN_{theory}$ and $BSC_{theory}$ calculations to understand in detail how the size distribution
impacts the relationship between CCN and AOPs.

**Appendix A**

The AMS-measured ion concentrations of $NH_4^+$, $SO_4^{2-}$, and $NO_3^-$ must first be converted into volume fractions required by Eq.
(3). For this conversion, we first use the simplified ion pairing scheme developed by Gysel et al. (2007) to calculate the number
of moles (n) of ammonium nitrate ($NH_4NO_3$), sulfuric acid ($H_2SO_4$), ammonium bisulfate ($NH_4HSO_4$), and ammonium sulfate
(($NH_4)_2SO_4$), as outlined in Eq. (A1)-Eq. (A5),

$$n_{NH_4NO_3} = n_{NO_3^-}. \tag{A1}$$

$$n_{H_2SO_4} = \max(0, n_{SO_4^{2-}} - n_{NH_4^+} + n_{NO_3^-}) \tag{A2}$$

$$n_{NH_4HSO_4} = \min(2n_{SO_4^{2-}} - n_{NH_4^+} + n_{NO_3^-}, n_{NH_4^+} - n_{NO_3^-}) \tag{A3}$$

$$n_{(NH_4)_2SO_4} = \max(n_{NH_4^+} - n_{NO_3^-} - n_{SO_4^{2-}}, 0) \tag{A4}$$

$$n_{HNO_3} = 0, \tag{A5}$$

where the number of moles of $NH_4^+$, $SO_4^{2-}$, and $NO_3^-$ are calculated using their AMS-observed ion concentrations and molar
mass values. Next, the number of moles of ammonium nitrate, sulfuric acid, ammonium bisulfate, and ammonium sulfate are
converted to units of mass. After this step, their dry densities, as given in Table A1 (Gysel et al., 2007; Kuang et al., 2020),
are used to convert each mass to a volume. During this step, the AMS-measured concentration of organics is also converted to





a volume. The five resultant volumes are summed, and the total volume is used to calculate the volume fraction ($\varepsilon_i$) of each component. Following this step, the individual volume fractions and $\kappa_i$ values given in Table A1 (Cai et al., 2018; Kuang et al., 2020) are used in Eq. (3) to calculate a bulk kappa.


**Table A1: Density and hygroscopicity constants of individual chemical components used to calculate bulk hygroscopicity value.**

| Compound | $NH_4NO_3$ | $H_2SO_4$ | $NH_4HSO_4$ | $(NH_4)_2SO_4$ | Organics |
|---|---|---|---|---|---|
| Density $\rho$ (kg/m$^3$) | 1720 | 1830 | 1780 | 1769 | 1400 |
| $\kappa_i$ | 0.58 | 0.90 | 0.56 | 0.48 | 0.10 |

**Appendix B**

**Table B1: Dry refractive indices for each aerosol type. The two bottom rows represent the two combined aerosol types**
**used in this study. Their refractive indices are calculated using an average of both components from both aerosol types (i.e., the real and imaginary components for SFS are an average of the real and imaginary components for smoke and fresh smoke).**

| Aerosol Type | Real Component ($m_{dry}$) | Imaginary Component ($n_{dry}$) |
|---|---|---|
| Smoke | 1.505 | 2.005E-02 |
| Fresh Smoke | 1.425 | 2.005E-02 |
| Marine | 1.389 | 1.005E-03 |
| Polluted Marine | 1.407 | 5.050E-04 |
| Urban (URB) | 1.475 | 5.500E-03 |
| Smoke + Fresh Smoke (SFS) | 1.465 | 2.005E-02 |
| Marine + Polluted Marine (MPM) | 1.398 | 7.550E-04 |

The change in particle diameter is described using a hygroscopic growth factor g(RH), as defined in Eq. (B1),

$$g(RH) = \frac{D_{wet}(RH)}{D_{dry}}. \tag{B1}$$

Here $D_{dry}$ is the dry particle diameter from the SMPS and LAS observed size distribution, and $D_{wet}$ is the adjusted particle diameter at a given RH. To calculate $D_{wet}$, we follow the methodology of Zieger et al. (2013), who note that the RH dependence of Eq. (9) can be parameterized using a relationship introduced by Petters & Kreidenweis (2007), as given in Eq. (B2),



$$g(a_w) = \left(1 + \kappa \frac{a_w}{1 - a_w}\right)^{1/3},$$  (B2)

where $a_w$ is water activity and $\kappa$ is the bulk hygroscopicity parameter as calculated in Section 2.3.1. If the Kelvin effect can be neglected, $a_w$ can be replaced with RH. Since the Kelvin term of the Köhler equation is small for large particles (D > 80 nm), we make this replacement moving forward since particles larger than 80 nm contribute most to BSC compared to smaller particles. Therefore, we calculate humidified aerosol sizes using Eq. (B3),

$$D_{wet}(RH) = D_{dry}\left(1 + \kappa \frac{RH}{1 - RH}\right)^{1/3}.$$  (B3)

Additionally, the change of the refractive index due to hygroscopic growth is calculated using Eqs. (B4) and (B5) for the real ($m_{wet}$) and imaginary ($n_{wet}$) components, respectively,

$$m_{wet}(RH) = \frac{m_{dry} + m_{H_2O}(g(RH)^3 - 1)}{g(RH)^3},$$  (B4)

$$n_{wet}(RH) = \frac{n_{dry} + n_{H_2O}(g(RH)^3 - 1)}{g(RH)^3}.$$  (B5)

Here, $m_{dry}$ and $n_{dry}$ are the dry real and imaginary refractive indices for each aerosol type, as given in Table B1 and informed by Dubovik et al. (2002). Additionally, $m_{H2O}$ and $n_{H2O}$ are the real (1.33) and imaginary (0) refractive indices for water.

**Data Availability**

The HU-25 and King Air data are available through the NASA data archive:
https://doi.org/10.5067/SUBORBITAL/ACTIVATE/DATA001.

**Author Contributions**

EDL, LG, and JR formulated the observational and theoretical calculations studies. EDL organized all data products, performed analyses, visualized the results, and wrote the draft. LG, CAH, RAF, SPB, RHM, LDZ, EC, AS, CS, and JR edited the
manuscript and provided insightful discussion and suggestions.

**Competing Interests**

At least one of the (co-)authors is a member of the editorial board of Atmospheric Chemistry and Physics.

**Acknowledgements**

The ACTIVATE Earth Venture Suborbital-3 (EVS-3) investigation is funded by NASA's Earth Science Division and managed through the Earth System Science Pathfinder Program Office. EDL acknowledge support from NASA FINESST grant 80NSSC24K0008.





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
