# Peer review of "Aerosol effective radius governs the relationship between cloud condensation nuclei (CCN) concentration and aerosol backscatter"

_EGUsphere, 2025_

## Referee Comment (RC1)

Lenhardt et al. use in situ and HSRL measurements collected during the ACTIVATE campaign to investigate the factors controlling the relationship between cloud condensation nuclei (CCN) concentrations and aerosol backscatter coefficient (BSC). Given the current demand for a reliable global dataset of CCN concentrations, which have been attempted using satellite retrievals of aerosol optical properties such as BSC, their findings offer a valuable new perspective on remote sensing-based CCN retrievals. Particularly noteworthy is their demonstration of how aerosol size distribution (and thus effective radius) can vary significantly even within a given aerosol species, leading to a non-linear relationship between CCN and BSC. I found their methodology to be accurate and their interpretation of the results convincing. Uncertainties related to the study are also documented in detail. I recommend publication in *Atmospheric Chemistry and Physics* after the authors address the following (mostly minor) comments.

**Comments:**

**Line 140:** Consider discussing the observed order-of-magnitude variation in CCN concentrations for nearly constant BSC within a given aerosol type and relative humidity (RH). This would help frame the manuscript's central question regarding the drivers of such variability.

**Figure 4 (second column):** The label should read *o(Aerosol ID > 1)* since you exclude cases where more than one aerosol type is present.

**Figure 4 caption:** From the figure, it appears that only 3,939 out of 83,678 co-located samples are used in subsequent analyses. This significant filtering (about 95%) should be noted in the caption for clarity.

**Lines 380–382:** Please acknowledge that $BSC_{theory}$ is also derived using approximations. For example, assuming spherical particles in Mie theory and using climatological refractive indices for different aerosol types. These assumptions may contribute to the observed discrepancies.

**Sections 4.2 and 5.1:** The predictors used in the analysis are not fully independent. For instance, RH affects the effective radius ($R_{eff}$) depending on the aerosol's hygroscopicity. Therefore, the influence of RH on the CCN–BSC relationship may already be captured via $R_{eff}$. Moreover, $R_{eff}$ and geometric mean radius (*GMR*) are related through a well-defined expression if the aerosol size distribution follows a known functional form. For example, under a lognormal distribution, $R_{eff} = GMR \cdot exp\left(\frac{5}{2}\ln^2\sigma\right)$, where $\sigma$ is the geometric standard deviation. I suggest incorporating such relationships when interpreting the relative importance of the predictors.

**Section 4.2:** The low impact of Aerosol ID and κ on the CCN–BSC relationship is expected, since (i) BSC is primarily determined by aerosol size, and (ii) CCN activation is more sensitive to size than chemical composition (Dusek et al., 2006). I recommend including this discussion when presenting the relative importance of predictors.

**Line 12:** The abbreviation "ERFaci" is not used later in the abstract; consider removing or defining it where relevant.

**Line 68:** More recently, Choudhury et al. (2025) also reported a similar disagreement between aerosol extinction coefficient and CCN concentrations for marine aerosols across the globe.

**Line 133:** The abbreviation "URB" should be defined upon first use.

**Line 339:** Consider replacing "unnecessary" with "anomalous" for clarity.

**Line 495:** Revise to: "…steeper decrease in $CCN_{theory}:BSC_{theory}$ with $R_{eff}$"

**Line 496:** Correct the subscript format of $BSC_{theory}$; add "with $R_{eff}$" after $CCN_{theory}$

**References:**
Choudhury, G., Block, K., Haghighatnasab, M., Quaas, J., Goren, T., and Tesche, M.: Pristine oceans are a significant source of uncertainty in quantifying global cloud condensation nuclei, Atmos. Chem. Phys., 25, 3841–3856, https://doi.org/10.5194/acp-25-3841-2025, 2025.

Dusek, U., Frank, G.P., Hildebrandt, L., Curtius, J., Schneider, J., Walter, S., Chand, D., Drewnick, F., Hings, S., Jung, D. and Borrmann, S.: Size matters more than chemistry for cloud-nucleating ability of aerosol particles, *Science*, *312*(5778), 1375-1378, https://doi.org/10.1126/science.1125261, 2006.

---

## Author Comment (AC1)

**Response to Reviewer 1:**

We would like to thank the reviewer for the constructive feedback on our manuscript and for aiding our progress towards publication. These comments were very useful, and we appreciate the time taken to help improve the paper. Each comment is repeated here, and our responses are given below each one in blue text. Excerpts from the text of the paper are given in *italics*, where **new additions are bolded** and text removed is noted using . All line numbers mentioned in our responses correspond to the line numbers in the updated version of the manuscript.

Lenhardt et al. use in situ and HSRL measurements collected during the ACTIVATE campaign to investigate the factors controlling the relationship between cloud condensation nuclei (CCN) concentrations and aerosol backscatter coefficient (BSC). Given the current demand for a reliable global dataset of CCN concentrations, which have been attempted using satellite retrievals of aerosol optical properties such as BSC, their findings offer a valuable new perspective on remote sensing-based CCN retrievals. Particularly noteworthy is their demonstration of how aerosol size distribution (and thus effective radius) can vary significantly even within a given aerosol species, leading to a non linear relationship between CCN and BSC. I found their methodology to be accurate and their interpretation of the results convincing. Uncertainties related to the study are also documented in detail. I recommend publication in Atmospheric Chemistry and Physics after the authors address the following (mostly minor) comments.

Thank you for the constructive and positive feedback on our study!

**Line 140:** Consider discussing the observed order-of-magnitude variation in CCN concentrations for nearly constant BSC within a given aerosol type and relative humidity (RH). This would help frame the manuscript's central question regarding the drivers of such variability.

Thank you for this suggestion! We looked at the range of $CCN_{obs}$ corresponding to the peak of the $BSC_{obs}$ (0.0006-0.0008 km-1sr-1) and RH (80-90%) distributions and added the following sentences to Lines 144-148:

*"If we consider, as an example, the subset of SFS $CCN_{obs}$ with $BSC_{obs}$ between 0.0006-0.0008 $km^{-1}sr^{-1}$ and RH between 80-90%, both small ranges that capture the peak of observed conditions for SFS aerosols, $CCN_{obs}$ ranges from 25 to 2128 $cm^{-3}$. While this range captures the maximum observed variability, similar magnitudes can also be seen for MPM and URB aerosols within similar small ranges of $BSC_{obs}$ and RH."*

Additionally, to reiterate this point in the next paragraph when motivating the rest of the study, the following sentence has been added to Lines 160-161:

*"Additionally, we find that within individual aerosol types and for small ranges of $BSC_{obs}$ and ambient RH that the magnitude of $CCN_{obs}$ can vary by nearly two orders of magnitude."*

**Figure 4 (second column):** The label should read o(Aerosol ID > 1) since you exclude cases where more than one aerosol type is present.

Since the Aerosol ID defines aerosol types using arbitrary numbers, we exclude cases where more than one numeric value (aerosol type) is present. Therefore, we left this as σ(Aerosol ID) > 0 to signify that there is no variation in HSRL-2 defined aerosol types for a given data point.

**Figure 4 caption:** From the figure, it appears that only 3,939 out of 83,678 co-located samples are used in subsequent analyses. This significant filtering (about 95%) should be noted in the caption for clarity.

To clarify this significant filtering, the Figure 4 caption now reads:

*"Flowchart describing the data filtering steps applied to data for all analyses and filtering steps applied to data for specific analyses. The number of points remaining after each step (n) is given in parentheses.* **Therefore, approximately 25% of the original number of collocated samples remain for the observational analysis in Sect. 2, 5% for analyses comparing observations and theoretical values in Sect. 4.1, and 7% for purely theoretical analyses in Sect. 4.2 and 4.3."**

**Lines 380–382:** Please acknowledge that $BSC_{theory}$ is also derived using approximations. For example, assuming spherical particles in Mie theory and using climatological refractive indices for different aerosol types. These assumptions may contribute to the observed discrepancies.

To clarify and acknowledge the role of approximations in the BSC theory calculations, the following sentence was added to Lines 410-412:

*"Other discrepancies in the $BSC_{theory}$ calculation may come from approximations including the Mie theory assumption of spherical particles and our use of literature average refractive index values for different aerosol types."*

**Sections 4.2 and 5.1:** The predictors used in the analysis are not fully independent. For instance, RH affects the effective radius (Reff) depending on the aerosol's hygroscopicity. Therefore, the influence of RH on the CCN–BSC relationship may already be captured via Reff. Moreover, Reff and geometric mean radius (GMR) are related through a well-defined expression if the aerosol size distribution follows a known functional form. For example,

under a lognormal distribution, Reff = GMR · $ex\,p(5\,2\,\ln2\sigma)$, where $\sigma$ is the geometric standard deviation. I suggest incorporating such relationships when interpreting the relative importance of the predictors.

Thank you for pointing this out. We have added some discussion of these relationships in Sect. 4.2 (Lines 452-461), which now reads as:

*"It is important to note that the predictors used in this analysis are not fully independent. For example, RH impacts $R_{eff}$ depending on the corresponding kappa value, meaning that the influence of RH on the $CCN_{theory}$ – $BSC_{theory}$ relationship may be captured through $R_{eff}$. However, we include both parameters separately to investigate if one of these variables is more important than the other in constraining the $CCN_{theory}$ – $BSC_{theory}$ relationship. Additionally, both $R_{eff}$ and GMR capture the shape of the size distribution and can be related through functional relationships. We use $R_{eff}$ and GMR separately because of their different  information content. The weighting of $R_{eff}$ toward larger particles increases its relevance for AOPs, while GMR tends to fall within the fine mode of the size distribution closer to $D_{crit}$ and aerosol sizes relevant for CCN activation. Therefore, based on this combination of input variables we train the RF models to predict the ratio of $CCN_{theory}$:$BSC_{theory}$."*

An additional discussion of these relationships in terms of interpreting the predictor importance estimates was added in Sect. 5.1 (Lines 544-548), which reads as:

*"As previously mentioned, RH also has an impact on $R_{eff}$ that depends on kappa. The indication that $R_{eff}$ is the most important predictor suggests that understanding the $CCN_{theory}$:$BSC_{theory}$ relationship as based on ACTIVATE observations is not as straightforward as simply constraining RH, as could be done in L23. Rather, the impact of RH on the aerosol size distribution is more important in determining how $CCN_{theory}$ and $BSC_{theory}$ are related."*

**Section 4.2:** The low impact of Aerosol ID and κ on the CCN–BSC relationship is expected, since (i) BSC is primarily determined by aerosol size, and (ii) CCN activation is more sensitive to size than chemical composition (Dusek et al., 2006). I recommend including this discussion when presenting the relative importance of predictors.
The following sentence was added at the end of Sect. 4.2 (Lines 473-475):
*"The relatively low importance of Aerosol ID and kappa in these models is expected, considering $BSC_{theory}$ is primarily determined by aerosol size and CCN activation is also more sensitive to size than to aerosol chemical composition (Dusek et al., 2006)."*

**Line 12:** The abbreviation "ERFaci" is not used later in the abstract; consider removing or defining it where relevant.

The ERFaci abbreviation was removed from the abstract for clarity.

**Line 68:** More recently, Choudhury et al. (2025) also reported a similar disagreement between aerosol extinction coefficient and CCN concentrations for marine aerosols across the globe.
Thank you for making us aware of this additional paper! Choudhury et al. (2025) was added to the list of citations in Line 68 and to the reference list.

**Line 133:** The abbreviation "URB" should be defined upon first use.
The following sentence was added to Lines 126-127:
***"In this study, we combine smoke with fresh smoke (SFS) and marine with polluted marine (MPM) due to similarity in their optical properties. We also consider the urban/pollution (URB) aerosol type."***

Additionally, the sentence in Lines 206-207 was adjusted to read as:
***"As in Sect. 2, we*** *combine smoke with fresh smoke (SFS) and marine with polluted marine (MPM) due to similarity in their optical properties. We also consider the urban/pollution (URB) aerosol type."*

This is done to clarify that we use the same aerosol type abbreviations and combinations in the observational analysis and the theoretical calculation analysis.

**Line 339:** Consider replacing "unnecessary" with "anomalous" for clarity.
"Unnecessary" was changed to "anomalous."

**Line 495:** Revise to: "…steeper decrease in $CCN_{theory}:BSC_{theory}$ with $R_{eff}$"
This sentence was revised to include "with $R_{eff}$."

**Line 496:** Correct the subscript format of $BSC_{theory}$; add "with $R_{eff}$" after $CCN_{theory}$
The subscript was corrected and "with $R_{eff}$" was added.

---

## Author Comment (AC2)

**Response to Reviewer 2:**

We would like to thank the reviewer for the constructive feedback on our manuscript and for aiding our progress towards publication. These comments were very useful, and we appreciate the time taken to help improve the paper. Each comment is repeated here, and our responses are given below each one in blue text. Excerpts from the text of the paper are given in *italics*, where **new additions are bolded** and text removed is noted using . All line numbers mentioned in our responses correspond to the line numbers in the updated version of the manuscript.

The authors present a systematic observation- and modeling-based investigation of the relationship between the optical properties measured with lidar (particle backscatter coefficient, BSC) and the concentration of cloud condensation nuclei (CCN), i.e., of the number concentration of particles that can be activated to form cloud droplets. The paper is well written and a very good contribution to the lidar literature. The authors combined lidar observations of backscatter profiles with airborne in situ measurements of CCN. As a strong part of the manuscript, they included complex modeling of CCN and lidar backscatter coefficients. In this way, they clearly showed the dependence of the CCN-to-BSC conversion factor on the effective radius of the CCN.

A minor weak point is that they do not discuss and compare their effort with other methods. The use of conversion factors for different aerosol types (characterized by different effective radii) is indirectly the same approach than the one offered in the manuscript. Furthermore, the new approach is applicable to mixtures of hygroscopic particles (urban haze, marine particles, wild fires smoke), only! It is not applicable in the case of mixtures that include hydrophobic particles such as mineral dust. Then the different aerosol fractions have to be separated before a conversion into the dust and non-dust CCN fractions can be done. This point needs to be better addressed in the manuscript.

Thank you for the feedback on our study. As we will continue to discuss in response to some of the comments below, the point about lack of applicability of this method to mixtures with hydrophobic particles such as mineral dust is well-taken. We recognize that this study and the exact methods will not apply to dust aerosols, and this was not properly addressed in the original version of the manuscript. We have added a few places of discussion to the paper regarding this and will highlight them as they pertain to specific comments below.

In terms of a discussion/comparison of this effort with other methods, we are unaware of many other such studies that directly convert lidar observables to CCN concentration besides Haarig et al. (2019), as referenced in a comment below, and Lenhardt et al. (2023), which this study was largely motivated by. If there are others that use a conversion method as opposed to a physics-based retrieval or parameterization method, we would be happy to look into them further. Therefore, while we haven't addressed this type of approach specifically, a short paragraph was added to the beginning of Sect. 5 that acknowledges previous, related work using a variety of methods to get from lidar to CCN information and briefly discusses and summarizes the main differences in the work we have presented. This paragraph is in Lines 524-531 and reads as:

*"Several recent studies have used lidar observed aerosol optical properties to develop physics-based or ML (Machine Learning)-based parameterizations and retrieval methods for CCN concentration for different aerosol types (Mamouri & Ansmann, 2016; Lv et al., 2018; Haarig et al., 2019; Choudhury & Tesche, 2022a; Patel et al., 2024; Redemann & Gao, 2024). In this study, we have included in situ observed aerosol size and chemical composition information to determine which factors most strongly govern the $CCN_{theory}$ – $BSC_{theory}$ relationship.  Therefore, this analysis provides a broad theoretical context in which relationships between observed CCN and aerosol optical properties can be interpreted. In this section, we discuss the physical interpretation of the relationships found, implications for future remote sensing techniques, and a summary of the sources of uncertainty and limitations of the study."*

The manuscript is a methodology paper, and thus appropriate to AMT and should, to my opinion, be published in AMT, and not in ACP.

Respectfully, we do not agree with the assessment of our work as a methodology study – rather, the study aims to understand the underlying governing factors of the relationship between CCN and BSC from a theoretical perspective. In addition, we maintain that this manuscript is appropriate for ACP due to the broad applicability for future studies that will use lidar observables to retrieve CCN concentration. This is reflected by our statement in the abstract that the purpose of the theoretical calculations is *"understanding the dominant governing factors of the $CCN_{theory}$ – $BSC_{theory}$ relationship"* and why we end the abstract with a statement about how *"including information about aerosol size is critical for future studies in constraining CCN concentration from AOPs."*

Minor revisions are needed.

**Detailed comments:**

**Section 1** provides a good overview of CCN retrievals from optical measurements. One could even avoid a too broad discussion by focusing on profiling techniques.

Since our goal in future studies for a vertically resolved CCN product is to assess aerosol-cloud interactions, we decided to keep this information in the introduction. Additionally, we wanted to briefly discuss previous literature related to aerosol size and RH and how they impact both CCN and optical properties as an introduction to what we will focus on in the rest of the paper.

**Section 2:** Impressive field campaign! 'Unfortunately', the observed aerosol mixtures do not represent the full spectrum of relevant aerosol mixtures. The airborne HSRL-2, part of ACTIVATE, conducted many campaigns in the Caribbean and over the United States and detected mixtures of dust and non-dust components. One can find these relevant mixtures almost everywhere in the northern hemisphere, over all continents and adjacent oceans. However, this mixture is not covered in this study. This point needs to be better considered in the concluding discussion later on.

While we did not include HSRL-2 Aerosol ID identified observations of dust in this analysis, the observed aerosol types from ACTIVATE have been shown to be well-representative of mixtures of aerosol types observed in other campaigns (Fig S6 from Redemann & Gao, 2024), including in terms of dust components. Because of the statistical preponderance of what the HSRL-2 qualitatively identifies as smoke/fresh smoke, marine/polluted marine, and urban aerosol types, we focus on these types for our analysis. However, because of the qualitative nature of the HSRL typing technique, it is likely that there is a small amount of dust included in any of the aerosol types used in this study. Hence, inherently and to a certain extent, mixtures containing dust are likely included in our study. However, the reviewer's point about dust-*dominated* mixtures not being covered in the study is well taken. We have added some discussion to the final paragraph of Sect. 5.3 to 1) clarify that our results only hold for the aerosol mixtures observed in the ACTIVATE region and 2) speak to different considerations that would need to be made for mixtures including a larger proportion of dust. This adjustment is detailed in our response to the second to last comment in this document.

The wavelength of 532 nm is only mentioned in Section 2. It would be good to mention the wavelengths occasionally in the next sections (maybe also in some of the figure captions).

A mention of the 532 nm wavelength was added to the captions for Fig. 2 and 6. We also added brief mentions of the 532 nm wavelength in Lines 302, 314, 399, and 647.

P5, line 127: Please define R2!

In Lines 129-131 of the revised version, we now define $R^2$ (and RMSE) in the following sentence:

*"Additionally, we show* **the coefficient of determination ($R^2$), a measure of the proportion of variation in $CCN_{obs}$ that is explained by variation in $BSC_{obs}$,** *root mean square error (RMSE),* **a measure of the average difference between linear regression predicted CCN and $CCN_{obs}$,** *and number of data points (n)."*

P5, line 132: R is not introduced! You write: R2 values ranged from 0.0014 – 0.14 for all RH cases, and 0.0023-0.038 for RH<50%. I am confused! Such numbers indicate no correlation at all! What did I miss?

Apologies for the confusion - by discussing these $R^2$ values, the goal was to show that the correlations are weak (in contrast to our previous paper focusing on the ORACLES data set). To make this point clearer, Lines 135-137 have been adjusted to read as:

*"$R^2$ values for all aerosol types across the full RH spectrum range from 0.0014-0.14, and for RH ≤ 50% range from 0.0023-0.038,* **suggesting that there is no aerosol type for which variations in $CCN_{obs}$ are well-explained by changes in $BSC_{obs}$.** *For all RHs $R^2$ is strongest for URB, while smoke has the highest  $R^2$ under limited RH conditions."*

P 7, line 192: HSRL-2 Aerosol ID product! Is that defined somewhere? Please explain in a bit more detail!

Our definition for the HSRL-2 Aerosol ID product is now provided in Lines 201-206. To clarify that it is an additional HSRL-2 variable provided in the observed data set (and not a separate data set/product), the sentence in Lines 201-202 now reads as:

*"Additionally, since we are interested in the impact of different aerosol types on the CCN – BSC relationship, we also use the HSRL-2 Aerosol ID* ** variable from the observed data set."**

P8, line 195: You define eight, not just well-defined aerosol types! Afterwards you combine marine and polluted marine, aged smoke and fresh smoke, but you still have ice (is that an aerosol type?), dusty mix, and dust, and urban/pollution. What is the dusty mix? Later on, you do not consider DUST at all! Otherwise you would be in trouble with the 'simple' link between CCN/BSC and effective radius.

In the Aerosol ID typing algorithm defined in Burton et al. (2012), "dusty mix" is defined as "a general category that may include cases of dust mixed with a variety of other species." Ice refers to optically thin ice crystals/ice haze, such as that frequently observed during the ARCTAS campaign. To designate that ice isn't really an aerosol, we refer to these categories as "particle" types. However, it is true that we do not consider dust in this study. This is due in part to the differences mentioned by the reviewer for dust (such as hydrophobicity and

non-sphericity), but also largely because observations classified as dust/dusty mix by the HSRL-2 Aerosol ID only made up only about 9% of our HSRL-2 – in situ CCN collocated data points. We added a few sentences in Lines 208-213 to clarify which of these categories we do not use and the reasoning for focusing on smoke/fresh smoke, marine/polluted marine, and urban aerosols that we had a more significant amount of data for. It reads as follows:

*"These three aerosol types are the most frequently available in the ACTIVATE data. We do not consider observations categorized as ice, dusty mix, or dust in this study. Optically thin ice is infrequently detected by the HSRL-2 in ACTIVATE and does not designate an aerosol type relevant for CCN activation. Aerosols characterized as dust or dusty mix are also infrequently observed, making up only about 9% of the data points with a valid Aerosol ID, which does not permit a statistically relevant consideration of the dust-related aerosol types. Implications regarding the applicability of this analysis for dust contributions to aerosol mixtures will be discussed in Sect. 5.3."*

**Section 3.3:**

P 12, line 292: Any comment on volatile aerosol components? They are lost after drying and measuring/counting dry particles in situ. Humidification will not bring them and their impact back. They are not considered in... bin diameters and refractive index components.... Humidified bin diamters and refractive index components are the final input into the Mie scattering calculation runs in libRadtran.

We acknowledge this is an important source of uncertainty that we failed to discuss in the original paper and thank the reviewer for pointing it out. We added a brief discussion on the potential impact of volatile aerosol component loss in Sect. 5.3 (Lines 625-628):

*"Lastly, aerosols may be undersized due to loss of volatile aerosol components that occurs during the heating and drying of in situ observations during inlet transmission (Shrestha et al., 2018; Sandvik et al., 2019), and this may be another source of uncertainty in $BSC_{theory}$ and $CCN_{theory}$ calculations."*

**Section 4.1:**

Figure 5: The background values in the figure should be explained in the caption.
Thank you for catching this! The Fig. 5 caption now reads as:

*"$CCN_{obs}$ vs. $CCN_{theory}$ for (a) smoke and fresh smoke, (b) marine and polluted marine, and (c) urban aerosols. The 1:1 lines are dashed, and the lines of best fit for the linear regressions between both variables are solid. **Markers outlined in gray denote results for***

*calculations requiring a certain level of D$_{crit}$ agreement. Results for calculations not requiring a D$_{crit}$ agreement are shown in the background with lighter transparency to demonstrate how this requirement impacts the data set."*

Figure 5 shows the correlations for smoke, marine, and urban particles. For these aerosol types, your approach will work. As already mentioned above, in the case of mineral dust occurrence (hydrophobic particles with critical diameters around 200 nm) your approach would not work properly. You would have to use polarization lidar measurements to identify and separate the dust and non-dust contributions before estimating CCNC for the two particle fractions. The modelling part would need to consider the particle shape, which is still a big problem when it comes to BSC modeling (scattering phase function at 179.99 ° to 180°).

This aspect has to be discussed in the paper, may be at the end…

We agree with the reviewer that the difference for mineral dust is significant when it comes to the reliability of this method. We will address these considerations/differences in detail in Sect. 5.3 (changes outlined below in the final two comments). However, we would like to point out that the ACTIVATE dataset is unprecedented in its coverage of aerosol types in an airborne campaign, especially efforts that collocate remote sensing and in situ observations. The relative scarcity of ACTIVATE observations relevant for aerosol mixtures that include dust is unfortunate, but it does not detract from the novelty of analyses permitted for other aerosol types.

Equation 6: The effective radius of dry particles is given, should probably be mentioned again.

When rereading this section, we decided to clarify in a few additional places that we use the humidified size distributions for certain calculations (in addition to R$_{eff}$). To start, we specified in Eq. (5) and Lines 314-317 that we use humidified radius values as follows:

$$"BSC_{theory} = \int_{r_1}^{r_n} \pi r_{wet}^2 \, Q_{bsc} \, n(r_{wet}) dr_{wet}, \tag{5}$$

*where r$_{wet}$ is each humidified bin radius, n(r$_{wet}$)dr$_{wet}$ represents the aerosol number concentration in each bin, and r$_n$ represents the largest bin in the SMPS and LAS combined and humidified size distribution."*

Next, we made a few adjustments to clarify for Eq. (6) and (7) in Lines 446-450 that we calculate R$_{eff}$ and GMR for humidified aerosol size distributions as follows:

$$"R_{eff} = \frac{\int_0^\infty \pi r_{wet}^3 n(r_{wet}) dr_{wet}}{\int_0^\infty \pi r_{wet}^2 n(r_{wet}) dr_{wet}}, \tag{6}$$

*where $r_{wet}$ **is humidified** particle radius and $n(r_{wet})dr_{wet}$ is the aerosol concentration within each bin of the **humidified** size distribution. Geometric mean radius is the mean of the* **humidified** *aerosol size distribution in log space, as given by Eq. (7),*

$$GMR = \left( \frac{\int_0^\infty \ln r_{wet}\, n(r_{wet}) dr_{wet}}{N_0} \right),$$  *(7)"*

**Section 4.3**

Figure 8. Without the black exponential curve fit, one would hardly see any correlation. The uncertainty is high.

There is uncertainty/scatter in this figure due to 1) the theoretical calculations of CCN and BSC being based on in situ observed aerosol size distributions and 2) each size distribution being subject to humidification at 10 RH values for these calculations. However, based on our comparisons of calculated BSC and CCN in Figures 5 and 6, we conclude that this CCN$_{theory}$:BSC$_{theory}$ ratio is reasonable. To emphasize that most data is clustered around the fit line, and to make the relationship/correlation more visible, we made the marker colors correspond to the density of surrounding points. However, we realize that the RMSE value on this figure can be misleading/difficult to interpret based on the large magnitude of the CCN:BSC ratio. Therefore, to show a more straightforward measure of error and to correspond more directly with Figure 9, we added the mean relative error to Figure 8. These values range from 30-52%, suggesting that the uncertainty is relatively low and on par with that given in Figure 9. The adjusted Figure 8 is provided below:

[Figure]

Figure 9 corroborates the applicability of the CCN-BSC approach when considering aerosol type information, i.e., when considering the effective radius. All this works for a 'pure' non-dust aerosol mixtures.

Agreed - we address the lack of applicability for dust mixtures in Sect. 5.3 (changes outlined below in the final two comments).

**Section 5.2**

P 21, line 520: A simple linear approximation with BSCobs will not well constrain CCN obs in most cases. This 'general' statement holds for ACTIVATE aerosol mixtures. Other approaches try to find solutions to separate or isolate different aerosol types and then apply conversions. This was for example shown for Barbados dust/pollution mixtures (Haarig et al., 2019, references are given below).

We clarified that this general statement is true for ACTIVATE aerosol mixtures in Line 572: *"First, a simple linear approximation with $BSC_{obs}$ will not well-constrain $CCN_{obs}$ in most cases **in the ACTIVATE data set**."*

Additionally, we included the Haarig et al. (2019) reference in Line 53 within the Introduction.

**Section 5.3**

P 23, lines 565-577: One could add the original paper pointing to non spherical marine particles (Haarig et al., 2017). And regarding dust, the papers of Haarig et al. (2022) and of Saito and Ping paper (2021) provide an impression on the latest modeling approaches with focus on lidar products....

Thank you for making us aware of these additional studies. We have adjusted the sentence in Lines 619-620:

*"First, marine aerosols have a greater tendency compared to smoke and urban aerosols to be non-spherical in shape, as **was observed over Barbados by Haarig et al. (2017) and** has been discussed for the ACTIVATE dataset by Ferrare et al. (2023)…"*

Additionally, the final paragraph of Sect. 5.3 (Lines 633-642) was edited to incorporate a summary of the limitations related to the non-dust aerosol mixtures that we use for this study. We clarify that the method only applies to aerosol mixtures present over the ACTIVATE campaign region and speak to some potential difficulties of using this same method for observations of dust:

*"Lastly, there are a few important considerations for the applicability and limitations of this study. While the ACTIVATE campaign collected  **one of the most complete airborne datasets in terms of the range of aerosol types and meteorological conditions,** our findings are limited to the campaign study area and **the encountered aerosol mixtures;** they have not been tested on other datasets. **For example, since we**

*are unable to include dust in the analysis due to observational constraints, our results cannot speak to differences in the $CCN_{theory} - BSC_{theory}$ relationship for aerosol mixtures with large proportions of dust. We would expect the results shown here to differ for observations of dust due in part to its hydrophobic nature and large, generally non-spherical sizes and shapes not easily represented using Mie theory. Recent studies have started using lidar products to better model and understand dust aerosol optical properties (Saito & Ping, 2021; Haarig et al., 2022), but more work is needed to understand the relationship between dust optical properties and its ability to activate as CCN. Additionally, as previously mentioned, we would also expect the general exponential relationship between $CCN_{theory}:BSC_{theory} - R_{eff}$ to hold for other **non-dust** data sets, but the exact fit coefficients would likely need to be adjusted."*

**Section 6**

P 24, line 601: ..that Reff well captures the strong dependence of the CCN-BSC relationship on the aerosol size distribution..... YES, this is true for the ACTIVATE mixtures. Now, we need to broaden the spectrum towards dust/non-dust aerosol mixtures.

We edited the sentence in Lines 660-662 to include the following clarification:

*"Most importantly, we found through using a wide range of in situ observed size distributions that $R_{eff}$ well-captures the strong dependence of the $CCN_{theory} - BSC_{theory}$ relationship on the aerosol size distribution **for non-dust aerosol mixtures**."*